



# Influence of Relative Humidity on the Heterogeneous Oxidation of Secondary Organic Aerosol

**Ziyue Li[1,*], Katherine A. Smith[2], Christopher D. Cappa[2,*]**

[1]Atmospheric Sciences Graduate Group, University of California, Davis, USA

[2]Dept. of Civil and Environmental Engineering, University of California, Davis, USA

[*]Correspondence to: Ziyue Li (ziyli@ucdavis.edu) or Christopher D. Cappa (cdcappa@ucdavis.edu)

## Abstract

Secondary organic aerosol (SOA) is a complex mixture of hundreds of semi-volatile to extremely low-volatility organic compounds that are chemically processed in the atmosphere, including via heterogeneous oxidation by gas-phase radicals. Relative humidity (RH) has a substantial impact on particle phase, which can affect how SOA evolves in the atmosphere. In this study, SOA from dark $\alpha$-pinene ozonolysis is heterogeneously aged by OH radicals in a flowtube at low and high RH. At high RH (RH = 89%) there is substantial loss of particle volume (~60%) at an equivalent atmospheric OH exposure of 3 weeks. In contrast, at low RH (RH = 25%) there is little mass loss (<20%) at the same OH exposure. Mass spectra of the SOA particles were measured as a function of OH exposure using a vacuum ultraviolet aerosol mass spectrometer (VUV-AMS). The mass spectra observed at low RH overall exhibit minor changes with oxidation and negligible further changes above an OH exposure = 2 x $10^{12}$ molecule cm$^{-3}$ s, suggesting limited impact of oxidation on the particle composition. In contrast, the mass spectra observed at high RH exhibit substantial and continuous changes as a function of OH exposure. Further, at high RH clusters of peaks in the mass spectra exhibit unique decay patterns, suggesting different responses of various species to oxidation. A model of heterogeneous oxidation has been developed to understand the origin of the difference in aging between the low and high RH experiments. Differences in diffusivity of the SOA between the low and high RH experiments alone can explain the difference in compositional change but cannot explain the difference in mass loss. Instead, the difference in mass loss is attributable to RH-dependent differences in the OH uptake coefficient and/or the net probability of fragmentation, with either or both larger at high RH compared to low



RH. These results illustrate the important impact of relative humidity on the fate of SOA in the atmosphere.

## 1. Introduction

Organic aerosol (OA) is ubiquitous in the atmosphere and makes up a large fraction of
submicron particulate matter (Zhang et al., 2007). OA includes primary organic aerosol (POA) particles that are emitted directly into the atmosphere and secondary organic aerosol (SOA), which is formed in the atmosphere through oxidation of volatile organic compounds (VOCs) and yields less-volatile products that partition to the particle phase (Hallquist et al., 2009). SOA is often found to dominate OA mass (Jimenez et al., 2009), and thus can have a large
impact on the ability of particles to affect climate and air quality. Despite the importance of SOA, understanding its full lifecycle—involving formation, chemical evolution and removal processes—remains challenging due to its complex and variable composition (Hallquist et al., 2009).

Heterogeneous oxidation is an important process during the lifecycle of OA in the
atmosphere. Heterogeneous oxidation of OA involves the reaction of gas-phase oxidants, such as OH and $O_3$, with compounds in the condensed phase. Heterogeneous oxidation can alter the composition of organic particles through two general pathways, functionalization and fragmentation (Kroll et al., 2009;Donahue et al., 2012;Lambe et al., 2012;Nah et al., 2013;Wiegel et al., 2015). Functionalization leads to formation of more oxygenated species in
the particles (Smith et al., 2009;Ruehl et al., 2013) while fragmentation generally leads to production of more volatile species (Slade and Knopf, 2013). Heterogeneous oxidation can alter the optical properties (Cappa et al., 2011;Dennis-Smither et al., 2012), hygroscopicity (Petters et al., 2006;George et al., 2009;Cappa et al., 2011;Dennis-Smither et al., 2012;Harmon et al., 2013;Slade et al., 2015), density (George et al., 2007;Kroll et al., 2009;Nah
et al., 2014), thermal properties (Emanuelsson et al., 2014;Hu et al., 2016) and the particle size (George et al., 2009;Dennis-Smither et al., 2012;Donahue et al., 2012;Kroll et al., 2015) of OA .

Various laboratory experiments have considered heterogeneous oxidation by OH of OA particles or films of OA proxies; these have typically focused on single compound systems (e.g.
George et al., 2007;McNeill et al., 2008;Hennigan et al., 2010;Nah et al., 2013;Slade and Knopf, 2013;Chan et al., 2014;Davies and Wilson, 2015;Fan et al., 2015;Kroll et al., 2015;Lai et al.,





2015). In general, heterogeneous oxidation leads to an increase in the bulk-average oxygen-to-carbon ratio (or the average carbon oxidation state) as a function of photochemical age (Kroll et al., 2015). Heterogeneous oxidation by OH can also lead to varying amounts of OA mass loss, from negligible to more than 30% for OH exposures equivalent to 1 weeks of

typical ambient OH (Kroll et al., 2015).

One important consideration in understanding the impact of heterogeneous oxidation on OA properties is the influence of relative humidity and particle-phase water. The phase of OA particles depends on the OA composition and the relative humidity (Koop et al., 2011;Renbaum-Wolff et al., 2013). For SOA in general, particles at low RH tend to have very

high viscosity, with the particles behaving as if solid or semi-solid, although the exact behavior depends on the SOA precursor gas, oxidant and concentration (Virtanen et al., 2010). At high RH, however, the viscosity of SOA is generally much lower, with the particle phase being more liquid-like (Bateman et al., 2015). Diffusion coefficients ($D_{org}$) of molecules within particles are related to viscosity according to Stokes-Einstein relation, with lower diffusion coefficients

corresponding to higher viscosity. The molecular diffusivity is further related to the mixing timescale ($\tau$) of the particle. The relationship between $D_{org}$ and $\tau$ is approximately $\tau = d_p^2/4\pi^2 D_{org}$, where $d_p$ is the particle diameter. For reference, the $\tau$ of a 100-nm particle is ~2.5 seconds when $D_{org} = 1 \times 10^{-12}$ cm$^2$ s$^{-1}$, corresponding to a more "liquid-like" particle, while $\tau$ is ~40 min when $D_{org} = 1 \times 10^{-15}$ cm$^2$ s$^{-1}$, typical of "semi-solid" particles. Because RH impacts $D_{org}$

and mixing timescales, variations in RH have the potential to impact particle reactivity (Kuwata and Martin, 2012;Li et al., 2015b;Wang et al., 2015) and thus the evolution of particles in the atmosphere.

To address this, recent experimental studies have considered the effect of RH on heterogeneous oxidation by OH radicals, typically using model (single-component) OA

systems. Some studies have observed slower loss of the model compound (Chan et al., 2014;Davies and Wilson, 2015;Fan et al., 2015;Chim et al., 2017), or smaller gas-phase uptake coefficients (Slade and Knopf, 2014) under lower RH conditions. A number of model simulations have suggested the smaller OH uptake coefficient and longer photochemical lifetime of OA under dry conditions are driven by limited diffusion in highly viscous particles

(Arangio et al., 2015;Houle et al., 2015). However, other single-component studies report no effect of RH, or even a decrease in the loss of parent species and OH uptake (Park et al., 2008;Lai et al., 2014;Slade and Knopf, 2014), with suggestions that the existence of a water



film at the surface hinders uptake of gas-phase oxidants. Given these contrasting results, it is difficult to extrapolate from these model systems to understand how RH might influence the reactivity of chemically complex SOA. We are aware of only one study that has probed the influence of RH on reactivity of SOA. Hu et al. (2016) examined the dependence on RH of

heterogeneous oxidation of ambient isoprene-epoxydiol-derived (IEPOX) SOA by OH over the RH range 40 – 100%. They observed mass loss of IEPOX-SOA upon heterogeneous oxidation at all RH investigated. The extent of IEPOX-SOA loss was independent of RH for RH < 90%. However, at RH > 90% they found that the loss was somewhat enhanced, most likely as a result of an increase in particle surface area and not as a result of changes to particle phase

because viscosity of the IEPOX-SOA particles was likely sufficiently low that the particles exhibited liquid-like behavior at all RH conditions considered. Thus, the specific influence of SOA phase on heterogeneous reactivity remains unclear.

       To address this gap, we have conducted heterogeneous OH oxidation experiments under low and high RH conditions using SOA formed from dark ozonolysis of $\alpha$-pinene. We

selected $\alpha$-pinene SOA because oxidation of monoterpenes, including $\alpha$-pinene, is an important source of SOA globally (Guenther et al., 1995;Ziemann and Atkinson, 2012;Guenther et al., 2012). In addition, the composition, phase state and photochemical aging of $\alpha$-pinene SOA have been extensively investigated, although the impact of phase on photochemical aging has not been characterized. Here we establish the effect of RH on

photochemical aging of the $\alpha$-pinene SOA in terms of the influence on aerosol mass and compositional change. A simplified multiphase oxidation model that couples reaction and diffusion is used to understand the RH-dependent factors that impact photochemical aging.

## 2. Methods

       In the experiments here, SOA is formed initially under dry conditions and these SOA

particles are then heterogeneously oxidized by OH at either a low or high RH. The experimental setup is shown in **Figure 1**. Four parts are highlighted: SOA formation, RH control, heterogeneous oxidation and SOA measurements. SOA particles were first produced chemically in one flow tube under dry conditions (the SOA formation flow tube). The airstream was then humidified to the desired level, and the SOA particles were reacted

heterogeneously with OH radicals in a second flow tube (the OH flow tube). The size



distribution and chemical composition of the particles were then characterized. Each of these steps is described below.

### 2.1 SOA formation

SOA was formed from homogeneous nucleation of products from dark ozonolysis of $\alpha$-pinene in a stainless steel flowtube reactor in the absence of seed particles and $NO_x$. $\alpha$-pinene (98%, Sigma-Aldrich) was constantly injected into a stream of clean, dry $N_2$ (at 1.0 $\mu$l hr$^{-1}$) using two parallel syringe pumps. $O_3$ was produced by passing the $N_2/O_2$ mixture through a cell lit by Hg pen-ray lamp (UVP, LLC). These two flows were introduced into a 25 cm long, 1.2 cm diameter stainless steel tube containing two helical static mixers (StaMixCo,

LLC) and then passed into a 170 cm long, 2.5 cm diameter flow tube reactor. This step-wise increase in diameter facilitated mixing of the reactants and limited dead volume in the main flow tube. The typical residence time of the flow tube was around 80 seconds. The $O_3$ concentration was measured using an $O_3$ monitor (Model 108L, 2B Technologies) while the $\alpha$-pinene concentration is estimated based on flow rates and infusion rate. No OH scavenger

was used during SOA formation. Detailed information of each experiment is summarized in **Table 1**. Downstream of the flowtube was a Carulite 200 (Carus) denuder followed by a Charcoal denuder to remove residual hydrocarbons and oxidants in the gas-phase.

As shown in **Table 1**, mass concentrations of SOA were ~1000 $\mu$g m$^{-3}$, which is substantially higher than typical ambient aerosol concentrations, by two orders of magnitude

(depending on the environment). However, the high initial concentrations are needed to allow for robust chemical characterization of the particles by the VUV-AMS at the much lower concentrations encountered after particles have lost much of their mass at high OH exposures.

### 2.2 RH control

Particles were reacted with gas-phase OH radicals in a second flow tube at two distinct

RH conditions: high RH and low RH. Song et al. (Song et al., 2016) summarize a variety of experimental observations of the influence of RH on the phase state of $\alpha$-pinene SOA. Although there are discrepancies in the exact values of $D_{org}$ for $\alpha$-pinene SOA in the literature, the observations generally indicate that $D_{org} < 10^{-14}$ cm$^2$ s$^{-1}$ below RH = 30% and $D_{org} > 10^{-12}$ cm$^2$ s$^{-1}$ above RH = 70%. Based on this, the experiments here were conducted at RH = 25% for

low RH experiments and at RH = 89% for high RH experiments. A dry $N_2$ stream, a $N_2$ stream





humidified by a water bubbler and the stream containing the SOA particles were combined after the denuders. Three steps were used to achieve the desired RH range in the OH flow tube. First, the fraction of dry and humidified $N_2$ was varied, with no dry $N_2$ for high RH experiments. Second, the SOA particle stream was passed through a Nafion humidifier (Perma

Pure, LLC) for high RH experiments, but through a bypass line for low RH experiments. Third, the humidifier was heated to slightly above ambient temperatures (to ~35 – 36 °C) to increase the absolute water vapor concentration in the flow for high RH experiments. At this temperature and given the very short residence time in the humidifier, evaporation of SOA should be minimal (Kolesar et al., 2015). Condensation between the humidifier and the OH

flowtube was avoided by maintaining the connecting tubing at the same temperature as humidifier.

RH and temperature were monitored both at the inlet and outlet of the OH flow tube using Omega temperature controllers (CNi Series) and Newport humidity probes (iTHP-5), and temperature in the OH flow tube was measured by a thermocouple (5SC-TT-K-30-36, Omega).

Temperature in the OH flow tube was usually a few degrees higher than room temperature due to heating from the UV lamps, 26 – 27 °C compared to 24 – 25 °C, respectively. The temperature at the flow tube outlet was controlled to be equal as the temperature in the OH flow tube such that the RH measured at the outlet was representative of the RH in the OH flow tube.

**2.3 Heterogeneous oxidation**

The SOA was heterogeneously oxidized by OH radicals in a 130 cm long quartz flow tube. The residence time was 38 s at the flow rate of 1.0 L min$^{-1}$. Four continuous output 130 cm long Hg lamps ($\lambda$ = 254 nm; Sankyo Denki) were positioned outside of the quartz flowtube. OH radicals were generated by the following reactions:

$$O_3 + h\nu \rightarrow O(^1D) + O_2 \tag{R1}$$
$$O(^1D) + H_2O \rightarrow 2OH \tag{R2}$$

At a fixed condition (dry vs. wet), the concentration of OH was varied by changing the

input $O_3$ concentration (Figure S1). $O_3$ was generated using a corona discharger (Model Y, Ozone Services). The $O_3$ concentration generated can be varied from 20 – 1000 ppm, which





corresponds to 1 – 50 ppm after dilution into the OH flow tube. The $O_3$ concentration was measured by an $O_3$ monitor (Model 202M, 2B Technologies) before dilution into the flow tube.

To estimate the OH exposure in the flowtube, a tracer compound, acetone ($H_3CC(O)CH_3$), was added to the flow. Acetone reacts with OH along the length of the flow tube with a known

rate coefficient, $k_{Ace+OH}$. We used the average temperature-dependent rate constant from Wollenhaupt et al. (2000); and Gierczak et al. (2003). The observed decay of acetone was used to calculate the OH exposure:

$$\frac{\ln([H_3CC(O)CH_3]/[H_3CC(O)CH_3]_0)}{-k_{Ace+OH}} = \int_0^t [OH]dt = \langle OH \rangle_t \cdot t = OH\ exposure \tag{1}$$

where $[H_3CC(O)CH_3]_0$ and $[H_3CC(O)CH_3]$ are the initial and final concentrations of acetone, respectively, [OH] is the time-dependent concentration of OH, $\langle OH \rangle_t$ is the time-averaged concentration of OH, and $t$ is the residence time in the quartz flow tube, which is fixed at 38 seconds (Figure S1). The acetone concentration was measured using a gas chromatograph

(Model 8610c, SRI Instruments). A cylinder of 50 ppm acetone balanced in $N_2$ was prepared and used. After dilution, the initial concentration of acetone entering the OH flow tube was ~ 750 ppb.

Water vapor was removed from the airstream that exited the OH flow tube (after the RH was measured) by a diffusion dryer and any remaining gas-phase hydrocarbons and

oxidants were removed by a Carulite denuder.

### 2.4 SOA characterization

Both particle size and composition were characterized after the OH flow tube. The size distributions of the SOA particles were characterized by a scanning mobility particle sizer (SMPS; TSI Inc.), composed of a differential mobility analyzer (DMA; Model 3085) and a

condensation particle counter (CPC; Model 3772). The SOA volume concentration at each OH exposure was calculated from the volume-weighted distribution. These volume concentrations ($V_p$) were normalized by the measured particle number concentration to account for variations in particle losses that result from changes in particle size upon heterogeneous oxidation. By comparing the number-normalized $V_p$ values at a given OH

exposure to that at no OH exposure, the extent of bulk volume (mass) loss as a function of OH exposure is obtained.



The composition of the SOA particles was characterized using the time-of-flight (ToF) vacuum ultraviolet aerosol mass spectrometry (VUV-AMS) at beamline 9.0.2 at the Advanced Light Source (ALS) at Lawrence Berkeley National Laboratory. A detailed description of the instrument has been reported elsewhere (Gloaguen et al., 2006). Briefly, particles are

sampled into the instrument and focused into a particle beam through an aerodynamic lens. The particles impact onto a heater block that is maintained at 120 – 125 °C and evaporate. The evaporated molecules are photoionized using 10.5 eV radiation. The resulting ions are extracted and directed into a ToF mass spectrometer for detection. Each particle measurement is coupled with a background measurement. The background spectra are

measured by sampling air into the VUV-AMS through a particle filter before the particle measurements. All the mass spectra reported here are background corrected. While the resolution of the VUV-AMS is $m/\Delta m \sim 2000$, the mass spectra reported here are in unit mass resolution.

### 2.5 Multilayer Heterogeneous Oxidation Model

We developed a simplified multilayer heterogeneous oxidation model that is based on both the KM-SUB, KM-GAP model (Shiraiwa et al., 2010;Shiraiwa et al., 2012) and a stochastic model (Wiegel et al., 2015;Wiegel et al., 2017). The model is used to develop understanding of the factors that influence the particle bulk behavior and compositional change under different RH conditions. Our model treats particles as 2D spheres having a fixed-depth surface

layer and $n$ variable-depth, sub-surface bulk layers, as illustrated in **Figure 2**. The depths of all sub-surface layers are assumed equal. Diffusion and chemical reactions are coupled in the model to simulate evolution of SOA.

Reactions of gas-phase $HO_x$ (OH and $HO_2$) radicals with condensed-phase organic species are assumed to occur only in the surface layer. Unlike the KM-SUB model and the

stochastic model, the physical processes of adsorption and desorption of $HO_x$ radicals are not explicitly simulated in the model. Production of $HO_x$ radicals from reactions that occur within the condensed phase is also not considered. Therefore, reactions involving $HO_x$ radicals are assumed to happen only at the surface, and not in the sub-surface bulk layers. This is a reasonable assumption because $HO_x$ radicals are very reactive species and they likely

primarily react within 1 nm from the surface in organic aerosols (Arangio et al., 2015) with the probability of reaction falling off rapidly with distance into the particle (Houle et al., 2015).




The OH and HO₂ reaction rates are thus calculated by using a single uptake coefficient (γ) that characterizes the fraction of collisions that lead to reaction of HO$_x$ radicals with organic molecules at the surface, as illustrated by Eqn. (2) (Smith et al., 2009):

$$\frac{d[Y]}{dt} = -\gamma_{OH}^{Y} \cdot f_Y \cdot J_{coll} \cdot C_p \cdot A \qquad\qquad (2)$$

where Y is any reactive stable organic species in the surface layer that can react with OH, $\gamma_{OH}^{Y}$ is the uptake coefficient of OH radicals by species Y, $f_Y$ is the fraction of Y molecules in the surface layer, $C_p$ is the particle number concentration and $A$ is the particle surface area of

one particle. $J_{coll}$ is the OH radicals flux at the particle surface, which can be calculated by Eqn. (3):

$$J_{coll} = \bar{c} \cdot [OH]/4 \qquad\qquad (3)$$

where $\bar{c}$ is the mean speed of gas-phase OH and $[OH]$ is the concentration of gas-phase OH. The same calculations for reaction rates are applied to reactions involving gas-phase HO₂ radicals. The gas phase concentrations of HO$_x$ radicals ([OH] and [HO₂]) are assumed constant throughout the flow tube for a given OH exposure. At each OH exposure, the OH concentration is set equal to the time-averaged concentration as calculated from the

specified OH exposure divided by total reaction time (38 s). The concentration of HO₂ is assumed to be same as that of OH. Using the reaction scheme of Li et al. (2015a) to simulate the more detailed gas-phase chemical reactions, it was determined that this was a reasonable assumption.

Since HO$_x$ radicals are assumed to react at the surface, the depth of the surface layer is

set to 0.76 nm, corresponding to an effective molecular diameter of typical SOA components. This value is estimated assuming an average molecular weight of 175 g mole$^{-1}$ and density of 1.3 g cm$^{-3}$ for α-pinene SOA (Renbaum and Smith, 2009). The depth of the surface layer is assumed independent of the total particle size, i.e. is constant.

A generalized reaction scheme of OH oxidation of alkanes suggested by Ruehl et al.

(2013) and Wiegel et al. (2015) is used to simulate the compositional change of particles, as shown in Fig. 2b. It is assumed that all of the initial (unreacted) SOA molecules react identically.



It is also assumed that all functionalized stable products formed are reactive towards OH, and up to three generations of products are considered. The general reaction scheme for both SOA parent species and for stable product species (written as $P_x$, where x indicates the generation number, from 0 – 3) can be written as:

$$P_x + OH \xrightarrow{O_2} RO_2 \cdot \qquad\qquad (R3)$$

$$RO_2 \cdot + HO_2 \rightarrow ROOH + O_2 \qquad\qquad (R4)$$

$$RO_2 \cdot + RO_2 \cdot \rightarrow 2ROH_x \qquad\qquad (R5a)$$

$$RO_2 \cdot + RO_2 \cdot \rightarrow 2RO \cdot \qquad\qquad (R5b)$$

$$RO \cdot \rightarrow F_1 + F_2 \qquad\qquad (R6a)$$

$$RO \cdot + O_2 \rightarrow ROH_x \qquad\qquad (R6b)$$

$$RO \cdot + P_x \rightarrow ROH_x + RO_2 \cdot \qquad\qquad (R6c)$$

The stable product species that can go on to react are the ROOH and ROH species. The above
scheme illustrates that oxidation is initiated by H-abstraction from a stable species (parent or product) by OH (*R3*), leading to production of an organic peroxy radical ($RO_2 \cdot$). It is implicitly assumed that the alkyl radical that proceeds $RO_2 \cdot$ formation reacts rapidly with $O_2$. The $RO_2 \cdot$ radicals can then react with either $HO_2$ or $RO_2 \cdot$. Reaction with $HO_2$ (*R4*) leads to production of organic hydroperoxides (ROOH). Reaction of $RO_2 \cdot$ with another $RO_2 \cdot$ can lead to production of
either functionalized stable products (alcohols or ketones; *R5a*) or alkoxy radicals ($RO \cdot$; *R5b*). All stable product species produced from $RO_2 \cdot$ self-reactions are lumped into the generation-specific generic product species $ROH_x$, where x indicates the generation. The $RO \cdot$ can either react with $O_2$ (*R6b*) or other stable species (*R6c*) to produce stable products (ROH) or can decompose to produce two fragmentation products ($F_1$ and $F_2$; *R6a*). The
fragmentation products are assumed to have low molecular weight and high volatility, and thus evaporate from the particles leading to mass loss. The stable products can further react with OH radicals in the surface layer and lead to the next generation of products. Formation of ROOH and ROH is known as functionalization. Here, the net effect of R5 ($RO_2 \cdot + RO_2 \cdot$ reactions) and R6 (the fate of $RO \cdot$) is treated in a simplified manner. In particular, $RO \cdot$ is not
treated explicitly. Instead, the overall probability that $RO_2 \cdot + RO_2 \cdot$ ultimately leads to production of functionalized products or fragmentation products is considered as a lumped





probability ($p_{frag}$). The parameter $p_{frag}$ accounts for both the branching ratio of $RO_2\cdot$ + $RO_2\cdot$ reactions and the fractional loss of $RO\cdot$ from decomposition (*R6a*) versus functionalization (*R6b* and *R6c*). The overall reaction for $RO_2\cdot$ + $RO_2\cdot$ can then be written as:

$RO_2\cdot$ + $RO_2\cdot$ → 2 × (1−$p_{frag}$) × $ROH_x$ + 4 × $p_{frag}$ × F           (R7)

where F indicates all fragmentation products. The fragmentation products are assumed non-reactive towards OH and are the only species that can evaporate. This simplification of the reaction scheme serves to overcome lack of knowledge regarding the actual branching ratios

and rate coefficients associated with $RO\cdot$ decomposition and reaction with $O_2$ and other stable species. The value of $p_{frag}$ is not known *a priori*, but can be varied to establish the sensitivity of the model to functionalization versus fragmentation, and how this might differ depending on the RH (and particle phase state). It is assumed that $\gamma_{OH}$, $\gamma_{HO2}$, the rate coefficients for self-reaction of $RO_2\cdot$ radicals ($k_{RO2+RO2}$), and $p_{frag}$ are the same for all generations.

Generation-specific products are tracked through three generations of reaction. Subsequent reaction of 3rd generation species are assumed to reform 3rd generation species.

        Reactions occur within layers, and diffusion occurs between neighbor layers. The diffusion rates are calculated based on the equations used in the KM-SUB model (Shiraiwa et al., 2010). The diffusion coefficient ($D_{org}$) is assumed the same for parent compounds and

functionalization products (both stable products and radical intermediates), and are dependent upon the RH. The $D_{org}$ for fragments is assumed to be 1.4 times larger than for the other species to account for their smaller sizes. The model results are not sensitive to reasonable variations in $D_{org}$ of the fragments.

        Only fragmentation products are assumed to be sufficiently volatile to evaporate. All

other species are assumed effectively non-volatile. The evaporation of fragmentation products is treated in a simplified manner wherein once these compounds reach the surface layer (either because they are directly produced in that layer or diffuse up from lower layers), they evaporate instantaneously. The change of particle size is calculated based on loss of fragments assuming a linear relation between molecule number and particle volume. As

particles shrink, the thickness of the sub-surface bulk layers decreases based on the new particle size and constant total layer number, and the volume of each layer and molecular



density of each species in each layer are recalculated. Density and averaged molecular weight of the aerosol are assumed to be constant over the process of oxidation.

Both the particle volume and composition after 38 seconds (the residence time of the quartz flow tube) of reaction are calculated as a function of OH concentration. The evolution

of the particle volume and composition with OH exposure are then determined from the individual model results at each OH concentration. The simulated volume fraction remaining (VFR) of particles as a function of OH exposure can be directly and quantitatively compared to the observed VFR while the simulated compositional changes can be qualitatively compared to the measurement of spectral change.

Water uptake affects the SOA in various ways. Water uptake can engender a change (decrease) in the SOA viscosity, cause the particle size to increase with a corresponding dilution of the organic molecules, impact the uptake of OH (i.e. $\gamma$), or alter the oxidation chemistry in the condensed phase (i.e. $p_{frag}$). The impact of viscosity changes are considered in the model by systematically varying the $D_{org}$ while assuming a constant, RH-independent

size, composition, $\gamma$ and $p_{frag}$ (Section 3.3.1). The impact of the size increase and dilution are considered together by using an increased particle surface area and a decreased molecular density of organic compounds for high RH simulations, where the extent of growth, characterized by the hygroscopic growth factor ($GF$) is constrained by literature observations (Section 3.3.2). In these simulations, the hygroscopicity of the products is assumed the same

as the parent species, and thus the assumed growth factor is constant with oxidation. The impact of variations in either $\gamma$ or $p_{frag}$ between low and high RH conditions are considered together (Section 3.3.3). These parameters are tuned one at a time for low and high RH conditions (using RH-dependent $D_{org}$ and size) to achieve good agreement with the measurements.

## 3. Results and Discussion

### 3.1 Bulk Behavior

The heterogeneous oxidation of $\alpha$-pinene SOA by OH leads to mass loss and particle shrinking for both the low and high RH condition. However, the extent of shrinkage at a given OH exposure differs between low and high RH (**Figure 3**). An example of the change in the

size distributions before and after heterogeneous oxidation at the same OH exposure (= 4.7





x $10^{12}$ molecule cm$^{-3}$ s) is shown for both low and high RH (**Figure 3**b). The general broadening

of the size distribution after exposure to OH is due to the slower overall reaction for larger

particles resulting from their smaller surface-to-volume ratio. The variation in the volume loss

with OH exposure between low and high RH is shown in **Figure 3**a. At an OH exposure = 4.7 x

$10^{12}$ molecule cm$^{-3}$ s, only 20% of particle volume (or mass assuming constant particle density)

is lost at low RH, compared to more than 60% volume loss at high RH. These observations

clearly indicate that particles shrink to a much greater extent under high RH. These

observations have implications for the lifetime of SOA with respect to heterogeneous

oxidation in the atmosphere. Assuming a typical atmospheric [OH] of 2 x $10^6$ molecules cm$^{-3}$,

it would take 4 weeks for $\alpha$-pinene SOA to lose 20% of volume under dry conditions but only

a few days to lose the same amount of volume under wet conditions. These observations

suggest that SOA mass loss due to heterogeneous OH oxidation may vary substantially

throughout the atmosphere, dependent on variations in ambient RH.

Our observations for SOA can be compared to various single-component observations

from the literature. Fan et al. (2015) reported the mass fraction remaining for methyl

β-D-glucopyranoside (a cellulose oligomer surrogate) nanoparticles after photochemical

aging by OH radicals at RH = 10 – 30%. The extent of mass loss was observed to increase with

RH to a small extent in this range. Chan et al. (2014) have also observed that aqueous (high

RH) succinic acid particles lost volume to a greater extent than solid (low RH) succinic acid

particles at the same OH exposure. However, they observed no further change in volume

above OH exposure ~ 0.8 x $10^{12}$ molecule cm$^{-3}$ s for aqueous droplets, different from our

observation of continuous volume loss at high RH. This is likely a result of the different

chemical systems used. Davies and Wilson (2015) reported the change in diameter of citric

acid particles as a function of OH lifetimes for RH= 20 – 90%. For all RH conditions, they

observed that the particle diameter initially increased but then peaked and decreased as the

number of oxidation lifetimes was increased. Their oxidation lifetime values can be converted

to equivalent, RH-dependent OH exposures using the reported rate constant. We find that

there is a greater change in particle diameter for a given OH exposure for their experiments

when RH is higher, with greater overall volume loss at high RH. Additionally, the initial, small

increase in diameter occurs at relatively low OH exposures (< 1.5 x $10^{12}$ molecules s cm$^{-3}$), and

the general effects of RH on particle volume loss with OH exposure are reasonably consistent

with our observations. In addition to the single component studies, our results can also be



compared to the observations of Hu et al. (2016), who reported the mass fraction remaining for IEPOX-SOA in the ambient environment as a function of OH exposure. As noted in the introduction, they observed faster decay of particle mass at 90 – 100 % RH compared to < 90 % RH, but no further variation with RH at <90%. However, they also observed a larger surface

area concentration under 90 – 100 % RH, due to particle growth, which was postulated to facilitate faster OH uptake. An increase in surface area is unlikely to explain our observation because the water uptake of $\alpha$-pinene SOA at RH = 90% is small (Varutbangkul et al., 2006) and the dry particle size distributions of our non-oxidized particles are nearly identical between the two RH conditions (Fig. 3b). Further, the increase in surface area is offset by a

decrease in the mole fraction of the organic species due to the presence of water, which will be examined in Section 3.3.2.

### 3.2 Spectral Changes

The composition of SOA generated from ozonolysis of $\alpha$-pinene has been well-studied using both online and offline mass spectrometry (e.g. Tolocka et al., 2006;Shilling et al.,

2009;Camredon et al., 2010;Putman et al., 2012;Yasmeen et al., 2012;Hall et al., 2013;Kristensen et al., 2014;Kristensen et al., 2016;Zhang et al., 2017). The mass spectrum obtained depends on the measurement method used. For example, mass spectra of $\alpha$-pinene SOA obtained using high-temperature evaporation (> 600 °C) coupled with electron impact ionization at 70 eV (the typical operating conditions of the Aerodyne AMS) have very few ions

with $m/z$ > 100 having substantial intensity (Shilling et al., 2009). In contrast, mass spectra obtained using electrospray ionization (ESI), typically considered a softer ionization method, typically show that dimers or oligomers (nominally those compounds having molecular weights more than twice higher than that of parent compound, 136 amu for $\alpha$-pinene) comprise a substantial fraction of the particle (Kristensen et al., 2014;Zhang et al., 2017). Mass

spectra for $\alpha$-pinene SOA obtained using the VUV-AMS, used here, exhibit peaks spanning the range $m/z$ = 50 – 300, although with much lower intensity ions at $m/z$ > 200 amu (**Figure 4**). In the VUV-AMS spectra, ions in range $m/z$ = 50 – 200 are likely a combination of monomers (especially above $m/z$ = 140), fragments of monomers (especially below $m/z$ = 140) and fragments of dimers. For example, characteristic fragment ions of monomers such as

cis-pinonic acid and pinic acid, known products of $\alpha$-pinene ozonolysis (Yu et al., 1999;Kristensen et al., 2014), have been previously identified in VUV-AMS mass spectra of



non-oxidized particles (Mysak, 2006) and are observed here with high intensity in the non-oxidized particles. The characteristic ions are observed at $m/z$ = 98, 114, 125, 166 for cis-pinonic acid ($m/z$ = 184) and at $m/z$ = 100, 114, 140, 168 for pinic acid ($m/z$ = 186), respectively. However, some of the odd-numbered, high-intensity ions that were observed in the

non-oxidized particles, such as $m/z$ = 125, 141 and 169 (**Figure 4**), may be fragments of dimers based on Kristensen et al. (2016), although they used a different mass spectrometry technique (ESI). This is consistent with the peaks in the VUV-AMS spectrum of $\alpha$-pinene SOA consisting of contributions of monomers, fragments of monomers and fragments of dimers. Dimers and oligomers are likely underrepresented in VUV-AMS spectra of $\alpha$-pinene SOA,

although they may comprise more than 50% of the aerosol mass (Kristensen et al., 2016). This underrepresentation is probably attributable to substantial fragmentation of dimers during the ionization process and/or the slow desorption of dimers from the heater block, leading to low sensitivity of VUV-AMS towards dimers and oligomers.

      The relative changes of the $\alpha$-pinene SOA mass spectra are considered as a function of

OH exposure. We define non-oxidized particles as particles that passed through the OH flowtube with the lights on but with no added O₃. As such, all changes to the observed spectra are attributable to chemical oxidation of the particles, not photolysis. We note that there is no difference in the mass spectra for non-oxidized particles for tests done with the lights on versus lights off in the absence of $O_3$ for low RH conditions ($R^2$ = 0.997; Figure S2). This

indicates that direct photolysis of the SOA had no influence on the SOA spectra for the short exposure timescale (38 s) here. We also observe that the mass spectrum of non-oxidized particles at low and high RH are identical ($R^2$ = 0.999), indicating that photolysis does not differ between low and high RH conditions in these experiments. Thus, the differences in spectral changes between the two RH conditions are driven by chemical oxidation only.

The mass spectra observed at four different OH exposures are shown for low RH (**Figure 4**a) and high RH (**Figure 4**b). There are distinct RH-dependent differences in the spectral evolution. At low RH, the four most abundant ions in the non-oxidized particle spectra ($m/z$ = 98, 125, 141 and 169), are still among the most abundant at the highest OH exposure. In contrast, at high RH the relative signal intensities of these four ions decreases substantially

with increasing OH exposure.

      The overall similarity of the oxidized spectra to the non-oxidized spectra can be considered by calculating the $R^2$ between the spectra (Lanz et al., 2007). Smaller $R^2$ values



indicate greater spectral differences (lower similarity). The $R^2$ between the mass spectra of oxidized particles and non-oxidized particles are calculated at each OH exposure for both the low and high RH experiments. Only ions that contribute more than 0.5% to the total signals are included in the calculation. As shown in **Figure 5**, changes in the $R^2$ differ substantially

between low and high RH. At low RH, the $R^2$ decreases reasonably fast below and OH exposure = 2 x $10^{12}$ molecule cm$^{-3}$ s, although to a limited total extent, only to $R^2$ = 0.78. Above this value, further decreases in $R^2$ are small. In contrast, at high RH the $R^2$ exhibits a rapid decrease below OH exposure = 1 x $10^{12}$ molecule cm$^{-3}$ s to near zero. The very small $R^2$ values at high RH indicates that the composition of the oxidized particles differs substantially from the non-

oxidized particles.

The decay of each individual $m/z$ ion in the mass spectrum (from $m/z$ = 15 – 350) with OH exposure is shown in **Figure 6**. The intensity of each ion has been first normalized by the particle number concentration to account for changes in intensity due to variations in physical loss of particles in the flow tube, which likely result from changes in the particle size

distribution with OH exposure. The number-normalized intensities were then normalized by the observed intensity at zero OH exposure to assess the relative change. This resulting normalized intensities are also referred to here as the signal fraction remaining ($SFR$). The decay of the average $SFR$ (calculated from the individual $SFR$ curves) and the weighted-average $SFR$ (equal to total signal fraction remaining) are also shown in **Figure 6**. The

difference in the average $SFR$ decay between low and high RH conditions and the absolute magnitude of decay are consistent with the observed volume loss. However, the weighted-average $SFR$ decays to a greater extent than the average $SFR$ decay or mass loss under both low and high RH conditions. Ideally, loss of total signal should be equivalent to loss of particle mass if detection sensitivity towards all the species are the same. One possible explanation

of this difference is that the VUV-AMS exhibits a lower sensitivity towards product species compared to parent species. Another possibility is the underrepresentation of dimers and oligomers in VUV-AMS spectra, as mentioned above. If the latter is true, it also indicates that dimers should decay more slowly than monomers to account for the difference between mass loss and total signal loss.

There is substantially greater spread in the $SFR$ between individual ions at high RH compared to low RH at a given OH exposure. In other words, at low RH the intensities of the individual ions all decay to a similar extent. However, at high RH the intensity of some ions



decrease by nearly a factor of 100 at the highest OH exposure while the intensity of others remains similar to the non-oxidized particles. This is consistent with the differences in $R^2$ versus OH exposure between low and high RH, indicating distinctly different compositional change.

We focus first on the high RH ion decay curves. To further understand differences between ions that decay rapidly versus slowly under high RH conditions, the ions have been grouped by their extent of decay (**Figure 7**a). Four groups have been established. Group 1 includes ions that exhibit the fastest and most substantial decay, with the average *SFR* for this group equal to 0.04 at the highest OH exposure. This group contains most of the ions that

have the greatest signal intensity in the non-oxidized particle spectrum and includes markers from known products of α-pinene ozonolysis, such as cis-pinonic acid and pinic acid. Also, the odd ions with *m/z* = 125, 141, 169 that might be markers of oligomers as discussed above are also in Group 1. Nearly all of the ions in Group 1 have m/z < 200 (**Figure 7**b), suggesting that Group 1 primarily consists of either monomeric species or fragments thereof, and signature

fragments from dimers (or larger oligomers). That these ions exhibit a rapid, continuous decay indicates that they are chemically transformed, producing either new functionalized products or fragments that evaporated, and are not produced during heterogeneous oxidation.

Group 2 includes the ions that decay relatively slowly, albeit still to a substantial extent, with the average *SFR* = 0.3 for this group at the highest OH exposure. Group 2 has the greatest

number of ions, containing many ions with m/z < 200 and, notably, almost all the ions with m/z > 200 (**Figure 7**c). The Group 2 ions with m/z < 200 have generally smaller percentage contributions to the total signal in the non-oxidized particles than the Group 1 ions. This suggests that they are either less characteristic minor ions for major parent compounds or characteristic ions for minor parent compounds in the non-oxidized particles. The Group 2

ions with m/z > 200 are contributed by very highly oxygenated products, dimer fragments and dimers. The comparably slow decay of Group 2 ions indicates either that they react substantially more slowly than the Group 1 species (which are likely monomers) or that production offsets some of the chemical loss during the aging process. It is difficult to determine which is more likely.

Group 3 includes ions that exhibited little decay with oxidation. The Group 3 ions all have m/z < 50 (**Figure 7**d), and thus are small ion fragments that could result from any of the particle-phase compounds, both parent and product species. The relative intensity of the



Group 3 ions is overall extremely small. That their intensity remains relatively constant suggests that these ion fragments are produced to a greater extent from oxidation products than from the molecules comprising the non-oxidized SOA. In other words, production is offsetting loss. Group 4 includes the one ion ($m/z$ = 17) that exhibits an increase in signal

intensity. The relative intensity of this ion is very small, contributing only 0.01% to total signal in non-oxidized particles. However, at the highest OH exposure the percentage contribution of this ion increases to nearly 1%. Since these experiments were performed under $NO_x$ free conditions, $m/z$ = 17 most likely corresponds to $OH^+$ ions. In general, the formation of $OH^+$ ions from photoionization is unfavorable, which explains the extremely low intensity of this

ion in the mass spectra. However, the increase in the intensity of this ion with oxidation is likely still meaningful and could be a result of product species of increasing generation number having a greater number of -OH groups. Alternatively, it could indicate that the functional groups produced from oxidation are more likely to generate OH ions compared to the parent species, due to differences in bond dissociation energies. Most likely, the increase

in intensity of this ion is a result of functionalization.

         Considering now the low RH experiment (**Figure 6**a), all the ions visually appear to decay to similar extents. However, building on the results from the high RH experiment above, the mass spectrum has been split into the same $m/z$ groups (**Figure 8**). It is found that the Group 1 ions collectively decay to the greatest extent for the low RH experiment, consistent with the

high RH experiment. This provides additional evidence that Group 1 ions are mainly contributed from monomeric species that are chemically degraded and are not produced through heterogeneous oxidation. The Group 2 ions decay on average to a lesser extent than Group 1 ions at low RH, consistent with the high RH experiment. There is, however, a fair amount of variability in the individual ions for this group (mainly driven by low intensity ions).

The Group 3 ions exhibit, on average, slightly less decay than Group 2 ions, although with a fair amount of variability driven by the low ion intensities. The one Group 4 ion ($m/z$ = 17) may increase slightly with OH exposure, but the overall behavior is difficult to discern given the small change and low ion intensity.

         The mean *SFR* of Group 1 ions equals 0.4 at low RH compared to 0.05 at high RH at the

same high OH exposure (= 4.7 x $10^{12}$ molecule $cm^{-3}$ s). This difference is comparable to the observed differences in decay of parent ions for model (single-component) OA particles upon heterogeneous oxidation by OH under low versus high RH conditions (Chan et al., 2014;Davies





and Wilson, 2015). As discussed above, ions in Group 1 are likely representative of monomeric species in parent compounds. With the assumption that Group 1 ions are only lost through heterogeneous oxidation, and not formed, the mean signal decay of Group 1 ions could be considered as the lower limit of decay of "parent species" for $\alpha$-pinene SOA. The upper limit

of the reaction rate coefficient of $\alpha$-pinene SOA with OH, $k_{SOA+OH}$, can be estimated based on the signal loss rate of Group 1 ions by fitting the decay of the average *SFR* of Group 1 ions with an exponential function:

$$SFR = \exp(-k_{SOA+OH} \cdot OHexposure) \qquad (4)$$

The derived $k_{SOA+OH}$ is 2.3 x $10^{-13}$ cm$^3$ molecule$^{-1}$ s$^{-1}$ for low RH and ~1.3 x $10^{-12}$ cm$^3$ molecule$^{-1}$ s$^{-1}$ for high RH conditions. This difference in rate constants is similar to the difference reported by Davies and Wilson (2015) for citric acid and Hu et al. (2016) for IEPOX-SOA for a similar RH difference. The latter study reported an approximately linear relation between the rate

coefficient and the wet particle surface area for the RH conditions examined. However, as discussed above, $\alpha$-pinene SOA only takes up small amount of water at RH = 90%, leading to an increase in surface area of only ~20%, which cannot explain the more than 5 times higher rate constant under high RH.

Overall, it has been observed that substantially larger changes are observed at high RH

than at low RH both in terms of the extent of mass loss and the change in the particle composition with increasing OH exposure. It has also been observed that under low RH conditions the composition of SOA stops changing (based on consideration of the $R^2$) above certain oxidation level while mass loss continues slowly. In contrast, both compositional change and mass loss occur continuously and dramatically under high RH conditions.

### 3.3 Model simulation: Dependence on RH

The model is used here to establish the factors that primarily lead to the observed difference in heterogeneous oxidation under different RH in terms of both mass loss and compositional change. Heterogeneous oxidation of organic aerosol has several key steps, as discussed above and shown in **Figure 2**. These include sticking and uptake of gas-phase HO$_x$

radicals, chemical reactions of different species and diffusion of the species in the bulk. All of



the steps can be affected by RH, and in particular the influence of RH on viscosity plays important roles in these steps.

For the uptake process, there are a variety of reasons that the reactivity of $HO_x$ radicals may differ between low and high RH conditions. The existence of water at the surface might

lead to more efficient sticking of gas-phase $HO_x$ radicals due to enhanced hydrogen bonding to water (Chan et al., 2014). However, the greater abundance of water vapor at high RH also might decrease the available sites for adsorption of gas-phase $HO_x$ radicals due to competitive co-adsorption (Kaiser et al., 2011). After adsorbing/sticking to the surface, gas-phase $HO_x$ radicals are mobile on surfaces prior to reaction (Slade and Knopf, 2013). Since different sites

have different reactivity depending on their bonding environment (Kwok and Atkinson, 1995), the reaction probability of gas-phase $HO_x$ radicals with surface species can be affected by molecular orientation and the mobility of the radicals. The orientation of the SOA molecules can potentially be affected by RH. At low RH the SOA molecules are likely less mobile and their particular orientation and alignment at the surface relatively static compared to high RH

(Moog et al., 1982). It is possible that a difference in orientation and mobility of surface molecules between low and high RH affects the probability that an OH radical reacts versus desorbs after colliding with and sticking to the surface.

The viscosity of particles could also have an influence on the chemical pathways and products formed from $RO_2^{\cdot} + RO_2^{\cdot}$ reactions within the bulk by affecting either the (i) reaction

probability or (ii) branching ratio, or both. Denisov and Afanas'ev (2005) proposed that $RO_2^{\cdot} + RO_2^{\cdot}$ react to form an unstable tetroxide, which then decomposes to produce a ketone and an alcohol (the Russell mechanism), two ketones and $H_2O_2$ (the Bennett-Summers mechanism), or two $RO^{\cdot}$ radicals. They also indicated that the decomposition rate constant of the tetroxide intermediate decreases with an increase in viscosity of the surrounding phase. The products

formed from decomposition of the tetroxide depends on the structure of the transition state and, in the condensed phase, on further reaction of radical products within a solvent cage. The presence of a solvent cage and the stability of the cage can potentially influence the branching ratio towards production of $RO^{\cdot}$ radicals relative to stable, functionalized product species. This is important because $RO^{\cdot}$ radicals are the only species that lead to fragmentation.

Another impact of an RH-dependent viscosity is on the mixing timescales for condensed phase species. The mixing timescale for particles depends on the condensed phase diffusivity, which is related to viscosity. For example, for $D_{org} = 10^{-12}$ cm$^2$ s$^{-1}$, the mixing timescale for a



100 nm particle is about 2 seconds (Koop et al., 2011). Thus, the particle can be generally considered well-mixed and liquid-like. However, for $D_{org}$ = $10^{-15}$ cm$^2$ s$^{-1}$ the mixing timescale for a 100 nm particle is ~40 minutes. In this case, mixing is slow and the particle can be considered as having a more semi-solid phase. As discussed above, the diffusion coefficient

(and viscosity) of $\alpha$-pinene SOA vary continuously with RH. In general, at >70% RH, the diffusivity of $\alpha$-pinene SOA is large ($D_{org}$ > $10^{-12}$ cm$^2$ s$^{-1}$) while at low RH the diffusivity is comparably small ($D_{org}$ < $10^{-14}$ cm$^2$ s$^{-1}$). Given this, for our experiments it is expected that the particles at low RH (~30%) are semi-solid while at high RH (~90%) they are liquid-like. When the particles are more viscous (semi-solid), reaction of the organic radicals generated from

reaction with OH will be primarily constrained to surface layers. In contrast, when the particles are less viscous (liquid-like), reaction of radicals may occur throughout the entire particle. Therefore, there is reason to think that the impact of heterogeneous oxidation on both distribution of parent and product species in the particles and mass loss should depend on RH.

In the model simulations, the influence of RH on the above processes is examined by systematically varying five input parameters: (i) the growth factor ($GF$), (ii) the OH uptake coefficient ($\gamma$), (iii) bulk diffusivity ($D_{org}$), (iv) the RO$_2\cdot$ + RO$_2\cdot$ reaction rate constant ($k_{RO2+RO2}$), and (v) the combined branching ratio and fragmentation probability ($p_{frag}$). A summary of simulation parameters is shown in **Table 2**.

### 3.3.1   Impact of variations in bulk diffusivity

The ability of differences in the bulk diffusivity alone, and thus in the location of reactions, to explain the observed low versus high RH dependence is considered first. We perform simulations where $\gamma$, $k_{RO2+RO2}$, and $p_{frag}$ are all held constant while $D_{org}$ is varied over a range constrained by the literature, from $10^{-16}$ to $10^{-11}$ cm$^2$ s$^{-1}$ by factors of 10 (Song et al.,

2015). (The $\gamma$, $k_{RO2+RO2}$ and $p_{frag}$ were all specified to produce model results generally consistent with the observations.) For these simulations it is assumed that the size of the particles is independent of RH (discussed in Section 3.3.2). We find that substantial differences in the extent of mass loss between simulations with low $D_{org}$ (corresponding to low RH) and high $D_{org}$ (corresponding to high RH), as was observed, can only be achieved if

the $k_{RO2+RO2}$ is assumed to be < $10^{-21}$ cm$^3$ molecule$^{-1}$ s$^{-1}$, an unreasonably low value (**Figure 9**). When $k_{RO2+RO2}$ is this low, the RO$_2\cdot$ + HO$_2$ reaction (which leads to functionalization) becomes





competitive in the surface layer at low RH and at high OH (HO$_2$) exposure. This leads to a build-up for ROOH species in the surface layer and limits the fragmentation pathway. At high RH, the RO$_2$· rapidly diffuse away from the surface into the bulk, where they primarily react with each other and can produce RO· radicals that can subsequently decompose and the

fragments can evaporate. This leads to a suppression of fragmentation at low RH compared to high RH especially at higher OH (HO$_2$) exposure. However, the $k_{RO2+RO2}$ in this case is orders of magnitude lower than reported in the literature for various RO$_2$· + RO$_2$· reactions (Denisov and Afanas'ev, 2005).

When a more reasonable value of $k_{RO2+RO2}$ is used (1 x 10$^{-15}$ cm$^3$ molecule$^{-1}$ s$^{-1}$), only very

small differences in the calculated VFR versus OH exposure curves between the two RH conditions (i.e. when $D_{org}$ is varied) are found no matter the values of $\gamma$ and $p_{frag}$, assuming these are RH-independent (**Figure 10**a). For these calculations, the $\gamma$ and $p_{frag}$ are selected to provide good model/measurement agreement with the VFR versus OH exposure curve for high RH conditions. (A similar negligible dependence of the VFR on $D_{org}$ is predicted when $\gamma$

and $p_{frag}$ are selected to fit the low RH observations.) It should be noted that the particular $\gamma$ and $p_{frag}$ used here do not provide a unique solution; various combinations give similar results. Thus, we conclude that differences in $D_{org}$ alone cannot explain the differences between the low and high RH experiments in terms of the overall mass loss. Simulations also show that the overall reaction rate of RO$_2$· + RO$_2$· in the particle is at least three orders of magnitudes higher

than the rate of RO$_2$· + HO$_2$ regardless of the bulk diffusivity or $\gamma$ and $p_{frag}$ used. Even if we assume that [HO$_2$] = 10·[OH] instead of [HO$_2$] = [OH], as suggested by some oxidation flow reactor studies (Li et al., 2015a;Peng et al., 2015), loss of RO$_2$· by reaction with HO$_2$ is still negligible compared to loss by reaction with RO$_2$·. This result suggests that the uptake of HO$_2$ and formation of hydroperoxides are not particularly important during the heterogeneous

oxidation of the SOA system considered here. This agrees with a previous study by Lakey et al. (2016), who put an upper limit of 0.001 to the uptake coefficient of HO$_2$ by $\alpha$-pinene SOA.

However, although the VFR versus OH exposure curves using different $D_{org}$ are very similar, there are substantial differences in terms of the calculated compositional changes for the different $D_{org}$ considered (**Figure 10**b). In particular, the fractional contribution of model

parent species in the oxidized particles depends on $D_{org}$, with larger contributions at smaller $D_{org}$. For example, when $D_{org}$ = 1 x 10$^{-15}$ cm$^2$ s$^{-1}$ (low RH), parent compounds make up more





than 90% of the total molecules at highest OH exposure (**Figure 10**c). In contrast, when $D_{org}$ = $1 \times 10^{-11}$ cm$^2$ s$^{-1}$ (high RH), oxidation products make up nearly all of the particle at the highest OH exposure (**Figure 10**g). This is consistent with the observation of more dramatic spectral change occurs under high RH conditions when particles are expected to have low viscosity.

The reason for this results from differences in the rate of exchange between the bulk and surface at the different RH values. At high RH, parent molecules exchange rapidly between the surface and the bulk layers. Therefore, OH can access all of the parent molecules in the particle, leading to rapid change of particle composition. In contrast, at low RH the OH radicals cannot effectively access parent molecules in below-surface layers due to the high

viscosity/low diffusivity. In this case, the surface becomes highly oxidized while the bulk of the particle volume remains largely unoxidized, corresponding to minimal change of the spectra with OH exposure under dry conditions. Large differences in the overall particle composition between low and high RH conditions do not require additional differences in either $\gamma_{OH}$ or the fragmentation probability, unlike for volume loss. This is because of the

compensating effects of reaction location and oxidation extent of the products. Every time a stable species reacts with OH, there is a probability of fragmentation. Therefore, the net probability of fragmentation increases with higher generations of products, i.e. when species are more oxidized. At low RH, reactions are limited to the surface layer, leading to more highly oxidized products that have gone through more generations. While the net probability of

fragmentation is therefore increased for these surface-layer molecules, they also inhibit reaction with the bulk of the particle and will be, on average, more oxidized. Ultimately, at low RH the volume loss proceeds through layer-by-layer oxidation and evaporation of the surface. In contrast, at high RH OH radicals can react with the entirety of the particle bulk as these molecules mix to the surface. The average extent of oxidation at high RH is therefore

comparably lower than that in the surface-layer at low RH. However, a greater fraction of molecules have some probability of fragmenting and evaporating. The net effect is that the overall extent of volume loss is reasonably independent of RH and the particle bulk diffusivity, although clearly the reason for the volume loss (highly oxidized surface versus lightly oxidized bulk) depends on RH.

30        The model results also indicate that there is a threshold $D_{org}$ above which bulk compositional changes are large and below which they are small, with the threshold $D_{org} \sim$ $10^{-13}$ cm$^2$ s$^{-1}$. Above or below this value the model results are relatively insensitive to further




changes in $D_{org}$ (e.g. similar results are obtained for $D_{org}$ = $10^{-15}$ cm$^2$ s$^{-1}$ and $10^{-14}$ cm$^2$ s$^{-1}$, or for $D_{org}$ = $10^{-12}$ cm$^2$ s$^{-1}$ and $10^{-11}$ cm$^2$ s$^{-1}$). The particular threshold is related to the mixing time scale of the particles. For a 125 nm diameter particle (as used here), the mixing time scale when $D_{org}$ = $10^{-13}$ cm$^2$ s$^{-1}$ is about 40 seconds, which is comparable to the experimental time

scale. Thus, above $D_{org}$ = $10^{-13}$ cm$^2$ s$^{-1}$ the particles are effectively well-mixed and further increases in $D_{org}$ have limited influence, and at lower $D_{org}$ the particles do not mix.

### 3.3.2   Impact of size change

Besides affecting particle phase (i.e. $D_{org}$), uptake of water at high RH also leads to an

increase of particle size and dilution of organic compounds. The growth factor (*GF*) of $\alpha$-pinene SOA at RH = 90% is around 1.1 (Varutbangkul et al., 2006), corresponding to 21% and 33% increase in particle surface area and volume, respectively. The impact of this size increase is examined here by assuming that *GF* = 1.1 for high RH conditions and *GF* = 1.0 for low RH conditions, while all the other parameters are assumed RH independent. Multiple

simulations are run for different combinations of $\gamma$, $D_{org}$, $k_{RO2+RO2}$ and $p_{frag}$ (**Table 2**). We find that there are negligible differences between simulations that account for size growth (*GF* = 1.1) and those that do not (*GF* = 1.0) in terms of both the volume loss and compositional change of the particles for all combinations of the other parameters (i.e . $\gamma$, $D_{org}$, $k_{RO2+RO2}$ and $p_{frag}$) explored. Example results are shown in Figure S3. The reason for the insensitivity to *GF*

is that the increase in the surface area is offset by the decrease in the molecular density of organic compounds.

### 3.3.3   Influence of variations in OH uptake and condensed-phase reactions

Given that variations in $D_{org}$ alone cannot simultaneously explain the differences in observed mass loss and chemical changes at low versus high RH, coupled with the insight that

the difference in hygroscopic growth for low/high RH has a negligible impact on the simulations, the impact of varying either $\gamma$ or $p_{frag}$ with RH is considered. For all calculations that follow it is assumed that $D_{org}$ = $10^{-12}$ cm$^2$ s$^{-1}$ for high RH and $D_{org}$ = $10^{-14}$ cm$^2$ s$^{-1}$ for low RH, based on the discussion in the Section 3.3.1. It is also assumed that *GF* = 1.1 for high RH and that *GF* = 1.0 for low RH simulations. The model is successful in predicting bulk behavior, i.e.

mass loss of particles over the full range of OH exposure (= 0 − 7 x $10^{12}$ molecule cm$^{-3}$ s), when





RH-specific values of $D_{org}$ and $GF$ are used along with RH-specific values of either $\gamma$ or $p_{frag}$ (**Figure 11**a). For both conditions $k_{RO2+RO2} = 1 \times 10^{-15}$ cm$^3$ molecule$^{-1}$ s$^{-1}$ is used. For each condition, various combinations of $\gamma$ and $p_{frag}$ yield the same VFR versus OH exposure curves. As shown in **Figure 11**b, $\gamma$ and $p_{frag}$ have an inverse relationship, with increasing $\gamma$ accompanied

by decreasing $p_{frag}$ (to achieve the same fit to the observations). It is also observed that slopes of linear fits to log($\gamma$) versus log($p_{frag}$) for high and low RH conditions are identical, although the absolute values of $\gamma$ and $p_{frag}$ differ. This indicates that the relationship between either $\gamma_{dry}$ and $\gamma_{wet}$ or $p_{frag,dry}$ and $p_{frag,wet}$ required for good model-measurement agreement are independent of the absolute $\gamma$ and $p_{frag}$. In other words, when $p_{frag}$ is assumed

RH-independent, the $\gamma_{OH}$ always needs to be nearly five times faster at high RH conditions to reproduce the observed greater mass loss compared to low RH. Alternatively, when $\gamma$ is assumed RH-independent, the combined probability of fragmentation ($p_{frag}$) needs to be around four times higher at high RH to explain the difference in VFR between two conditions.

Although similar VFR versus OH exposure curves are obtained using various (RH-specific)

combinations of $\gamma$ and $p_{frag}$, the predicted variation in composition with OH exposure depends on the pair of $\gamma$ and $p_{frag}$ used. At high RH when $\gamma$ is small and $p_{frag}$ is large, the loss of parent molecules is relatively slow but with most reactions with OH leading to fragmentation and evaporation of the products. Consequently, there is little build-up of stable products in the particles (**Figure 11**c). In contrast, when $\gamma$ is large and $p_{frag}$ is small the parent compounds

react away faster but with a much larger fraction of stable products forming. Consequently, product species build up in the particles to a greater extent (**Figure 11**d). However, the net production rate of fragments in these two cases are the same, with the decrease in reaction rate offsetting the increase in fragmentation probability (or vice versa). The co-variation of $\gamma$ and $p_{frag}$ has a similar impact on the compositional changes for low RH conditions (**Figure**

**11**e,f), although the overall compositional changes are much smaller due to the high viscosity of the particles, as discussed above. Based on the above, we conclude that for high RH conditions the $\gamma$ must be greater than ca. 0.5 and $p_{frag}$ must be less than ca. 0.3 to achieve both substantial changes in the particle composition and substantial volatilization, as observed. If $\gamma_{wet}$ is assumed to be $\leqslant$ 1, then $p_{frag,wet}$ can be further constrained to be > 0.18.

Given that $\gamma_{wet} \sim 5\gamma_{dry}$ or $p_{frag,wet} \sim 4p_{frag,dry}$, as established above, the parameters at low RH are constrained to be 0.2 > $\gamma_{dry}$ > 0.1 and 0.075 > $p_{frag,dry}$ > 0.04.



### 3.4 Comparison with single-component studies

#### 3.4.1   OH Uptake

The results from our SOA experiments and model simulations can be compared with OH oxidation experiments performed on various single-component systems. Interestingly, some

of these studies find that increasing RH can enhance the effective uptake coefficient while others find that increasing RH reduces the effective uptake coefficient. The uptake coefficient can be calculated in two ways: (i) by measuring the loss rate of reacting particle-phase species, or (ii) by measuring the loss rate of gas-phase OH radicals. The first method may include loss due to secondary reactions in the condensed phase, in addition to direct loss from reaction

with OH, and is generally referred to as the effective OH uptake coefficient, $\gamma_{OH,eff}$. The latter method characterizes loss of OH only, and is referred to here as the OH uptake coefficient, $\gamma_{OH}$.

Some studies have observed a general increase in $\gamma_{OH}$ or $\gamma_{OH,eff}$ with RH when going from very dry conditions to higher RH ($40 - 70\%$).  For example, Davies and Wilson (2015) reported

that $\gamma_{OH,eff}$ for citric acid particles increases by a factor of 3 when RH is increased from 20% to 50%, although there is a slight decrease in $\gamma_{OH,eff}$ (by 30%) as RH is increased further to 90% RH. Chim et al. (2017) similarly observed a continuous increase in $\gamma_{OH,eff}$ from RH = 30% to 70% (from 1.9 to 2.6) and a small decrease at higher RH (to 2.4) for OH oxidation of particles composed of 2-methylglutaric acid. The increase in $\gamma_{OH,eff}$ with RH at the lower RH range was

explained by a decrease in viscosity and faster mixing. This allows OH radicals to react directly more often with the parent species, rather than producing highly oxidized molecules at the surface. The decrease of $\gamma_{OH,eff}$ at the higher RH range was explained, in part, by a decrease in the relative concentration of parent compounds due to dilution by water. Nevertheless, both studies indicate $\gamma_{OH,eff}$ is generally larger at high RH (~90%) than at low RH (~30%). Slade and

Knopf (2014) determined that $\gamma_{OH}$ (as opposed to $\gamma_{OH,eff}$) for levoglucosan particles increased with RH over the range 0 – 40%, from approximately 0.2 to 0.7. While the exact reason for this increase was not identified—although suggested to result from differences in phase state and mixing timescale—if the reaction products formed are less reactive towards OH than the levoglucosan, this could explain the increase in $\gamma_{OH}$ with RH because the products would build

up to a lesser extent at the surface when particle mixing is faster. Alternatively, it may be that




the reaction of OH with levoglucosan molecules at the particle surface depends on the orientation of the levoglucosan molecules. At very low RH the levoglucosan molecules may be fixed in unfavorable orientations (on average), and as RH is increased the levoglucosan may be able to adopt more favorable orientations. Finally, it may be that the sticking

probability of OH radicals increases with RH. Regardless of the exact reason, the above single-component studies are consistent with our determination that $k_{SOA+OH}$ for the Group 1 (parent) ions is greater at high RH.

However, some single-component studies have reported a negative effect of RH on the uptake coefficient or reaction rate constant, with increasing RH leading to smaller $\gamma$ or $k$. Slade

and Knopf (2014) observed a decrease in $\gamma_{OH}$ from 0.2 to 0.05 for methyl-nitrocatechol (MNC) particles when RH was increased from 0% to 30%. It was argued that because MNC has a low solubility in water the decrease was likely a result of the competition for adsorption between OH radicals and water at the surface; this differs from levoglucosan, which is moderately soluble. The SOA here is moderately soluble, due to the highly oxygenated nature of the

organic compounds, and thus most comparable to the levoglucosan experiments. Lai et al. (2014) reported a strong, negative effect of RH on the loss rate of levoglucosan after exposure to OH, in direct contrast to Slade and Knopf (2014). Lai et al. (2015) separately reported that the loss rate of $cis$-pinonic acid decreased, by a small amount, as RH increased from $20 - 80\%$. However, there is an important distinction between the Lai et al. (2015);(2014) and Slade and

Knopf (2014) experiments. Whereas Slade and Knopf (2014) used suspended particles, Lai et al. (2015);(2014) exposed thin films (<1 nm thickness) of the parent organic species to OH. Because the film was so thin the impact of diffusive exchange of molecules was effectively eliminated and the increase in RH served only to dilute the levoglucosan and decrease the loss rate. These results suggest that the experimental method used (e.g. suspended particles

versus thin films) can strongly impact the apparent influence of RH on heterogeneous oxidation processes. While experiments using suspended particles are more directly relevant to the atmosphere, the combination of thin film and suspended particle experiments can help to isolate the influence of mixing on reactive uptake and loss.

### 3.4.2 Reaction pathways

The influence of RH on the fate of the parent species from heterogeneous oxidation has also been considered by several studies. Following from faster loss of the parent species, Chan



et al. (2014) observed faster formation of both functionalization and fragmentation products for reaction of succinic acid in aqueous droplets compared to solid aerosol, i.e. at high versus low RH. As a result, they observed a more dramatic change in the mass spectrum after OH exposure under high RH conditions, consistent with our observation. However, the ions

formed after oxidation were identical for the aqueous and solid succinic acid particles. That is, while the product species form faster at high RH, there was no indication that this had a major influence on the reaction pathways. Chim et al. (2017) and Davies and Wilson (2015) similarly observed that, even though the reaction rates are different, the oxidation products formed are independent of RH. Furthermore, Chim et al. (2017) observed that for 2-MGA the

relative abundances of several important functionalization and fragmentation ions was independent of RH, when considered at the same 2-MGA lifetime (which accounts for RH variations), indicating similar chemical pathways of oxidation regardless of RH.

     In a chemical system containing carboxylic acids that are hydrophilic in nature, acid-base chemistry in the particles may need consideration in addition to free-radical chemistry when

an aqueous phase is present. Liu et al. (2017) developed a model that couples both acid-base and free-radical chemistry in the oxidation of aqueous citric acid aerosol by OH radicals. They compared their simulation results to the observations of Davies and Wilson (2015). The inclusion of acid-base chemistry in the simulations did not alter the decay rate of citric acid nor the variation in H/C and O/C as a function of OH exposure. However, a significant increase

in the abundance of fragmentation products were predicted only when acid-base chemistry is considered. One key reason is the enhanced unimolecular fragmentation rate of alkoxy radical anions compared to the neutral form. Hydration in the aqueous phase can also enable the decarboxylation of certain acyloxy radicals. Consistent with the simulations, Davies and Wilson (2015) reported that the abundance of $N_c = 6$ product species (the same carbon

number as citric acid) was smaller while the abundance of $N_c = 3 - 5$ product species (fragments or decarboxylation compounds) was larger at RH = 65% (corresponding liquid phase) compared to RH = 20%, although the differences were not significant. It is possible that acid-base chemistry, which we do not include in our simulations, contributes to the difference in oxidation between the low- and high-RH SOA experiments here. Considering

that acids can comprise a substantial fraction of $\alpha$-pinene SOA, it is possible that accelerated decomposition of alkoxy radicals in their anion form contributes to greater fragmentation under high RH conditions in the chemically complex system studied here. However, citric acid





is substantially more hygroscopic than $\alpha$-pinene SOA. The growth factor for citric acid at RH = 20% is similar to that of $\alpha$-pinene SOA at RH = 90%. As such, it is unclear to what extent acid-base chemistry might be playing a role in the oxidation of the $\alpha$-pinene SOA at RH = 90% and it may be negligible.

Regardless, RH-dependent differences in $p_{frag}$, and thus the oxidation products formed, can explain the difference in VFR and in the particle composition for SOA considered here based on our simulation results. Differences in viscosity of the particles has the potential to influence reaction rate and product branching ratio, and the importance of this may differ in the more chemically complex SOA compared to the single-component experiments.

Comparison with the single-component experiments suggests that the key difference between the low and high RH SOA experiments is more likely from the RH-dependence of $\gamma_{OH}$, although the role of variations in oxidation pathways ($p_{frag}$) cannot be completely ruled out.

## 4. Conclusions

The influence of RH on the heterogeneous oxidation of $\alpha$pinene SOA by gas-phase OH radicals is investigated for the first time. There is a substantial difference in the extent of

aerosol volume loss that results from photochemical aging between experiments conducted at low versus high RH (28% versus 90%), with much greater loss for high RH experiments. At low RH, the SOA particles are estimated to lose 80% of their volume over ~4 weeks of aging at typical atmospheric OH levels. In contrast, at high RH it takes only a few days for the same

amount of shrinking to occur. At high RH, substantial changes are observed in the mass spectrum of the SOA as the particles are oxidized. Nearly all peaks in the spectrum decrease in intensity as aging occurs, although to varying extents. Grouping of the peaks in the mass spectrum according to their individual extents of decay suggests that monomeric parent species exhibit the fastest decay. The decay of peaks that are attributable to dimers (or higher

order oligomers) is comparably slower, yet still substantial. Under low RH conditions, limited changes in the mass spectrum with oxidation are observed for the SOA particles. However, if the decay groups identified for the high RH experiment are applied to the low RH experiments, there is a general consistency between the low- and high-RH results, with slightly faster decay of the monomeric compounds compared to the dimer compounds.

The RH dependence of the observed volume loss and compositional change of the SOA particles was assessed using a multiphase chemical-oxidation model in which variations in the



bulk diffusivity, OH uptake coefficient and net probability of fragmentation were considered. Differences in the bulk diffusion coefficient alone, which affects mixing timescales within the particles, are found to have a negligible impact on the overall volume loss. However, the calculated variation in particle composition was highly sensitive to the bulk diffusion

coefficient and whether it was larger or smaller than a threshold value of $D_{org} \sim 10^{-13}$ cm$^2$ s$^{-1}$. At $D_{org}$ larger than the threshold value (faster mixing), substantial changes in the particle composition can occur as oxidation proceeds, whereas at $D_{org}$ smaller (slower mixing) than the threshold the particle composition remains approximately constant. The difference in volume loss between low and high RH are shown to result from either the OH uptake

coefficient or the net fragmentation probability (or both) being RH dependent, with larger values for high RH conditions. Considering the volume loss and compositional changes together, these two parameters are constrained to be $0.2 > \gamma_{dry} > 0.1$ and $0.075 > p_{frag,dry} > 0.04$ for low RH conditions and $1.0 > \gamma_{wet} > 0.5$ and $0.3 > p_{frag,wet} > 0.16$ for high RH conditions when $\gamma_{OH}$ is assumed to be less than 1. Comparison with studies of single compound systems

suggests that variations in $\gamma_{OH}$ is more likely responsible for the difference in low versus high RH experiments, although the possibility of different chemical pathways (i.e. differences in $p_{frag}$) due to difference in viscosity and the presence of water cannot be ruled out. Further work is necessary to establish which parameter, $\gamma$ or $p_{frag}$, is more sensitive to variations in RH, especially in chemically complex systems.

20       Overall, our work indicates that RH has a substantial impact on the evolution of the size and composition, and potentially related physical properties, of α-pinene SOA upon photochemical aging. Our observations suggest that the lifetime of SOA with respect to heterogeneous OH oxidation is significantly shorter at high RH and that this loss mechanism should be considered in regional and global models. Ultimately, additional studies with other

types of SOA (e.g. that formed from other biogenic or anthropogenic precursors) are needed to fully establish the effect of RH on SOA sensitivity to OH oxidation and to help further elucidate the underlying physical and chemical mechanisms. However, the results of the current study clearly point to the important role of RH in the heterogeneous oxidation of SOA.

## 5. Data Availability

30       The data associated with this paper have been submitted for archiving at the UC Davis DASH data repository (https://dash.ucdavis.edu) at https://doi.org/10.25338/B8ZK5X.





## 6. Acknowledgements

We thank Dr. Kevin Wilson, Michael Jacob and Nadja Heine at the Advanced Light Source for experimental assistance. Funding for this work was provided by the National Science Foundation (ATM-1151062). The Advanced Light Source is supported by the Director, Office

of Science, Office of Basic Energy Sciences, Chemical Sciences Division of the U.S. Department of Energy under Contract No. DE-AC02- 05CH1123. K.R.W. is supported by Department of Energy, Early Career Research Program, Office of Basic Energy Sciences, Chemical Sciences Division of the U.S. Department of Energy under Contract No. DE-AC02-05CH11231. This work was performed at Beamline 9.0.2 at the Advanced Light Source at Lawrence Berkeley National

Laboratory.

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


# Tables

**Table 1.** Experimental conditions for SOA generation and heterogeneous oxidation

|  | SOA precursors | | | SOA conditions | | | Experimental conditions | |
|---|---|---|---|---|---|---|---|---|
|  | $\alpha$-pinene ($\mu$l hr$^{-1}$) | $\alpha$-pinene (ppm) | O$_3$ (ppm) | $D_{p,S}$* (nm) | $N_p$ (# cm$^{-3}$) | $M_p$ ($\mu$g m$^{-3}$) | RH (%) | Humidifier T (°C) |
| Exp. 1 | 1.0 | 4.3 | 17 | 125 | 1.04e6 | 1120 | 89 | 35-36 |
| Exp. 2 | 1.0 | 4.3 | 17 | 126 | 1.12e6 | 1200 | 92 | 35-36 |
| Exp. 3 | 1.0 | 4.3 | 17 | 124 | 0.93e6 | 970 | 29 | 24-25 |
| Exp. 4 | 1.0 | 4.3 | 17 | 124 | 0.92e6 | 950 | 28 | 24-25 |

*$D_{p,S}$ is the surface-weighted middle diameter of initial dry particles

**Table 2.** Summary of values or ranges of values for parameters used in the model simulation

| Parameter | Description | Value |
|---|---|---|
| $D_{p,S}$ | Surface-weighted middle diameter of initial dry particles | 125 nm |
| $\rho_p$ | Density of SOA | 1.3 g cm$^{-3}$ |
| $MW_p$ | Molecular weight of SOA | 175 g mol$^{-1}$ |
| $GF$ | Hygroscopic growth factor | 1.1 (high RH) or 1.0 (low RH) |
| $\gamma$ | Uptake coefficient of OH and HO$_2$ | $0-1$ |
| $p_{frag}$ | A combined probability for fragmentation | $0-1$ |
| $k_{RO2+RO2}$ | Reaction rate coefficient of RO$_2^{\cdot}$ + RO$_2^{\cdot}$ | $1\times10^{-21} - 1\times10^{-15}$ cm$^3$ molecule$^{-1}$ s$^{-1}$ |
| $d_{SL}$ | Depth of surface layer | 0.76 nm |
| $D_{org}$ | Bulk diffusivity of SOA compound | $1\times10^{-16} - 1\times10^{-11}$ cm$^2$ s$^{-1}$ |
| [OH] | OH concentration along the flowtube | $0 - 1.84\times10^{11}$ molecule cm$^{-3}$-air |
| [HO$_2$] | HO$_2$ concentration along the flowtube | Same as [OH] |





# Figures

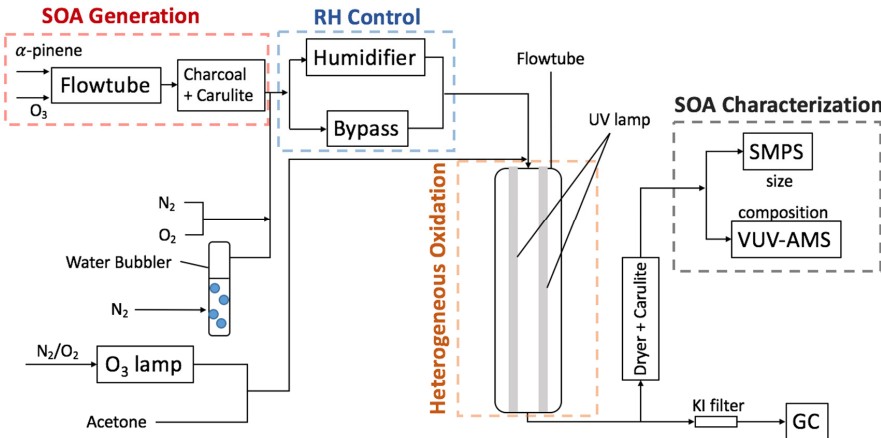

**Figure 1.** A schematic of the experimental setup for heterogeneous oxidation of SOA.





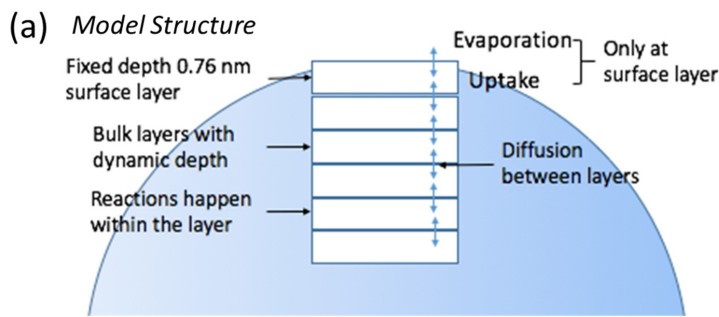

**Figure 2.** Overview of the simplified model. (a) The particle is treated as a sphere stacked with a fixed-depth surface layer and multiple bulk layers with dynamic depths that evolve as the particle evaporates. (b) A generalized reaction scheme of the oxidation process. Oxidation is initiated by H-abstraction of parent compounds that generate organic peroxy radicals ($RO_2^\cdot$), which react to form peroxides (ROOH), alcohols or ketones, or alkoxy radicals ($RO^\cdot$). $RO^\cdot$ radicals can react to form fragments or stable functionalized products. The net probability of fragmentation is treated implicitly in the model by combining the probability of alkoxy radical formation and their fragmentation. Uptake of HOx radicals and evaporation of fragments occurs only at the surface layer.




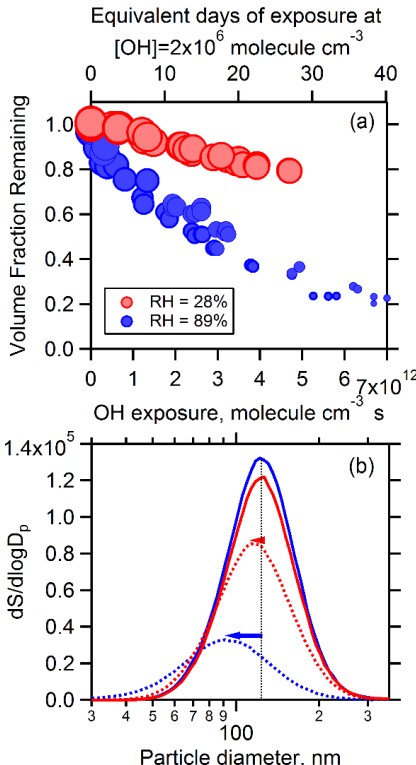

**Figure 3.** Influence of photochemical aging on SOA volume for low RH (red) or high RH (blue) conditions. (a) The volume fraction remaining as a function of OH exposure. Sizes of markers correspond to particle size. (b) Example surface-weighted size distributions of particle before OH exposure (solid lines) and after oxidation at an OH exposure = 4.7 x $10^{12}$ molecule $cm^{-3}$ s (dashed lines).



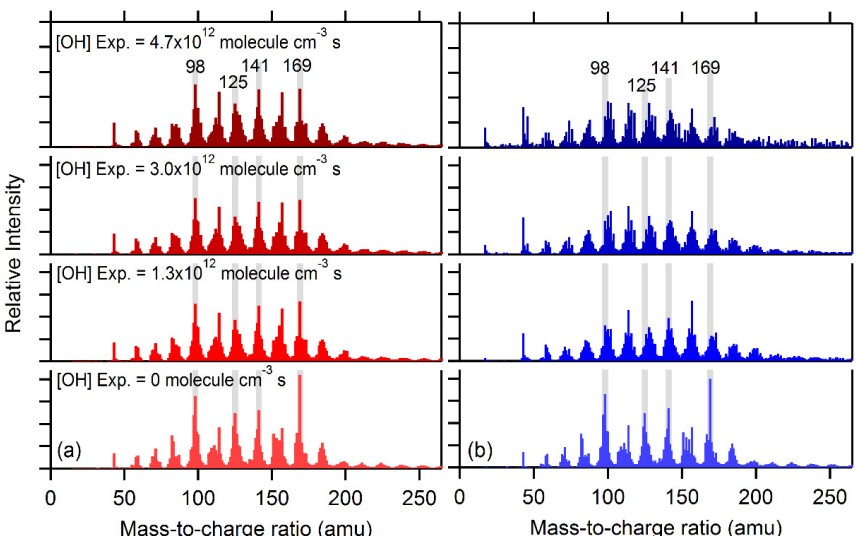

**Figure 4.** Example mass spectra of SOA as a function of OH exposure for (a) low RH (red, left panels) and (b) high RH (blue, right panels) conditions. OH exposure increases from the bottom to the top panels, with values listed in the panels. Vertical gray bars highlight four ions that exhibited a dramatic decay at high RH, but only small changes at low RH. The mass-to-charge ratio of these four ions are indicated in the figure.





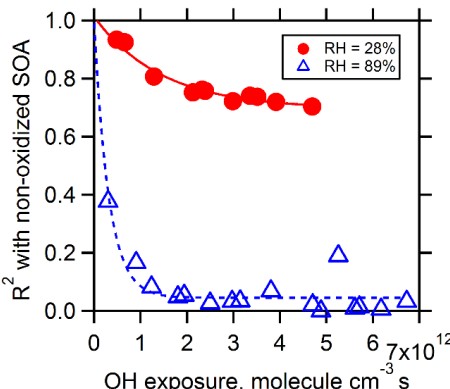

**Figure 5.** Coefficient of determination ($R^2$) between spectra of oxidized SOA and non-oxidized SOA as a function of OH exposure for dry (red circles) and wet (blue triangles) conditions. Spectra are filtered by excluding ions that have percentage contribution to total signal below 0.5 %. Lines are exponential fits, and presented only as visual guides.





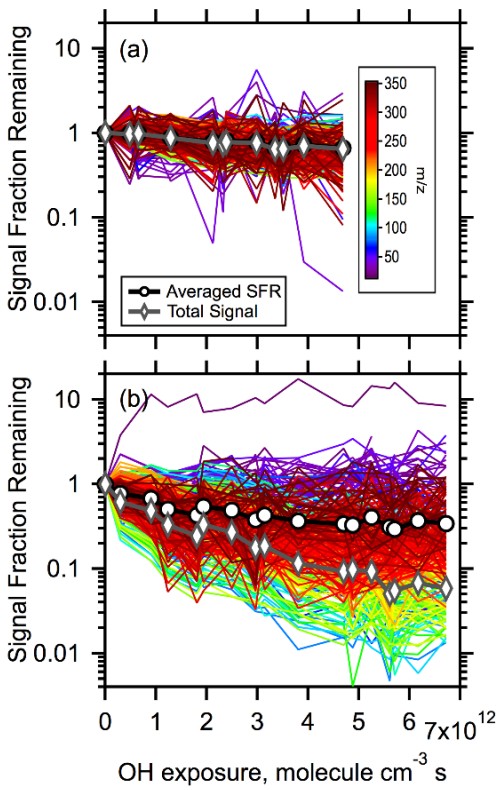

**Figure 6.** Signal decay of all peaks observed above background in the mass-to-charge range of 15 – 350 amu as a function of OH exposure for (a) low RH and (b) high RH conditions. Colors denote mass-to-charge of a given peak. The black open circles and line denotes the unweighted averaged decay of all the peaks. The dark gray open diamonds and line denote the total signal fraction remaining (i.e. the signal weighted average of all peaks). Note that log scale for the y-axis.



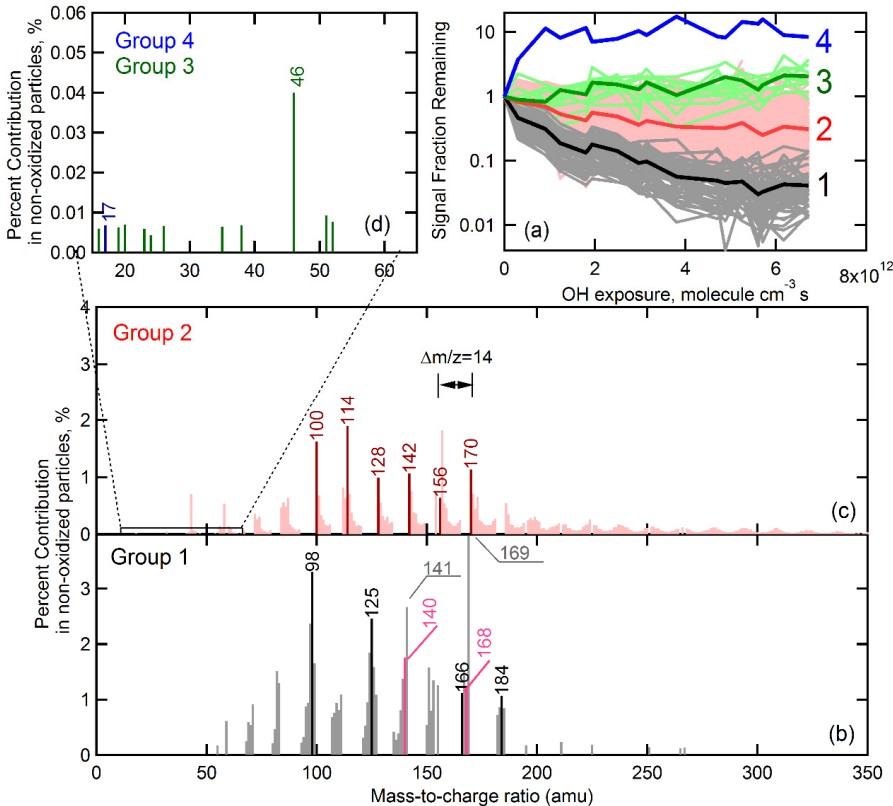

**Figure 7.** (a) Categorization of peaks according to their decay with OH exposure for high RH experiments. The peaks were classified into four groups. Peaks in Group 1 (gray) exhibit the fastest decay with the average shown as the solid black curve. The peaks in Group 2 (red) exhibit the second fastest decay, with the average shown in dark red curve. The peaks in Group 3 (green) exhibit negligible decay, with the average shown in dark green. Group 4 (blue) contains only one peak. (b) Spectrum of the peaks in Group 1. Group 1 contains markers from *cis*-pinonic acid (black peaks) and pinic acid (pink peaks). (c) Spectrum of the peaks in Group 2. Group 2 contains patterns of repeating peaks separated by $\Delta m/z = 14$, illustrated by dark red peaks. (d) A zoomed-in view of the peaks of Group 3 and Group 4. These peaks have very small intensities.





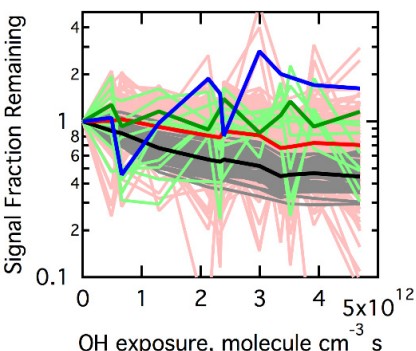

**Figure 8.** Signal fraction remaining of all the individual ions observed above background in the range m/z = 15 − 350 under low RH conditions. Ions are colored according to the groupings determined for the high RH conditions (c.f. **Figure 6**).




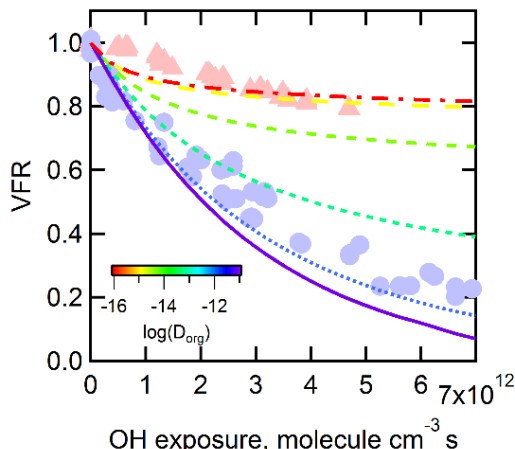

**Figure 9.** Comparison between observations (points) and simulations (lines) at varying $D_{org}$ of the dependence of the volume fraction remaining versus OH exposure for SOA for low RH (red triangles) and high RH (blue circles) conditions. The simulations assumed $\gamma = 0.6$, $p_{frag} = 0.5$, and $k_{RO2+RO2} = 3 \times 10^{-22}$ cm$^3$ molecule$^{-1}$ s$^{-1}$. The $D_{org}$ range from $10^{-16}$ to $10^{-11}$ cm$^2$ s$^{-1}$, denoted by colored lines.





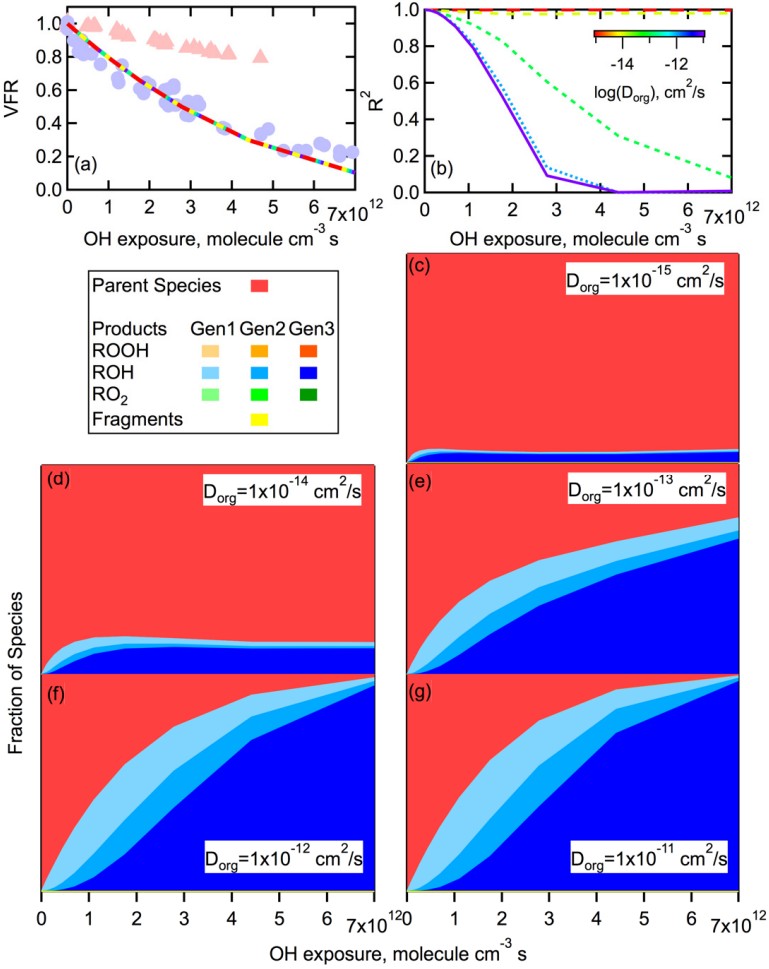

**Figure 10.** Simulated effect of variations in the diffusion coefficient on mass loss and compositional change. For these simulations, the $D_{org}$ is allowed to vary from $10^{-11}$ to $10^{-15}$ cm$^2$ s$^{-1}$ while all other parameters are held constant and chosen to give good agreement with the high RH observations ($\gamma$ = 0.50, $p_{frag}$ = 0.31). (a) The simulated volume fraction remaining versus OH exposure. Observations (symbols) are shown for reference for low RH (red triangles) and high RH (blue circles) conditions. The simulation results overlap because diffusivity has no effect on bulk mass loss. (b) The calculated coefficient of determination ($R^2$) between the molecular density of all simulated species as a function of OH exposure, referenced to the no oxidation case. The $D_{org}$ is indicated by the line color, with the purple solid line denoting the fastest diffusion. (c-g) Simulated fractional concentrations of each species as a function of OH exposure for different $D_{org}$. Colors indicate different species and generation (see legend). Only the ROH species are readily visible.


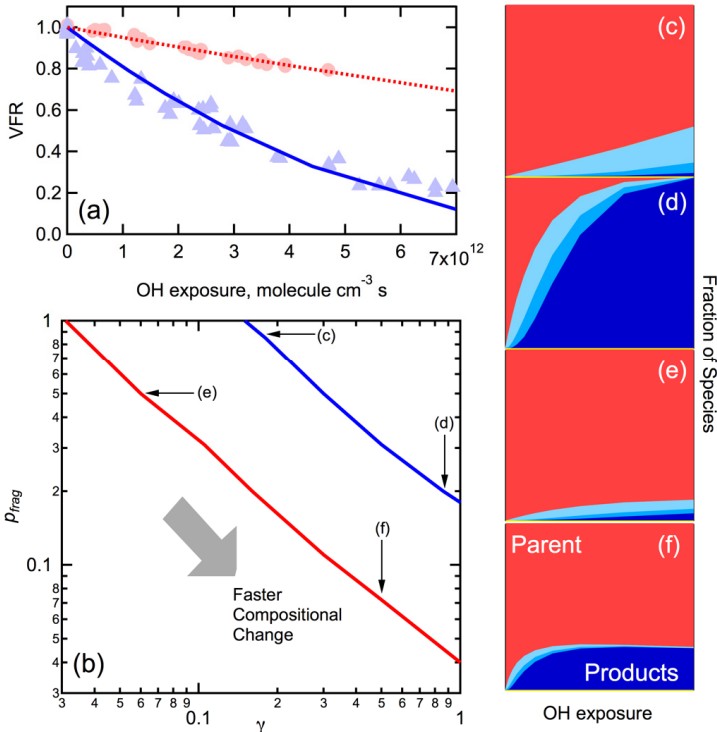

**Figure 11.** (a) Observed (points) and modeled (lines) VFR versus OH exposure for the best-fit models for both low RH (red) and high RH (blue) conditions. (b) Illustration of the relationship between $\gamma$ and $p_{frag}$ that allow for a good fit to the observed VFR decay for low RH (red) and high RH (blue) conditions. Four specific combinations of $\gamma$ and $p_{frag}$ are indicated for consideration of the associated composition change. (c-f) Simulated normalized composition change as a function of OH exposure for the $\gamma$ and $p_{frag}$ combinations indicated in panel (b). The colors correspond to different species with red indicating precursor (parent) species and other colors indicating various oxidation products. Greater compositional changes for a given OH exposure result from combinations with larger $\gamma$ and smaller $p_{frag}$ for a given RH condition.