# Peer review of "Influence of Relative Humidity on the Heterogeneous Oxidation of Secondary Organic Aerosol"

_Atmospheric Chemistry and Physics, 2018_

## Referee Comment (RC1) · Anonymous Referee #1 · 14 Jul 2018

Review of "influence of relative humidity on heterogeneous oxidation of secondary organic aerosol" by Li et al.

The authors explored the heterogeneous oxidation of secondary organic aerosols (SOA) by OH radicals as a function of relative humidity. A difference was observed between heterogeneous oxidation at 25% compared to 80%, and a model was developed to explain the results. This manuscript is easy to follow and clear. The science is also of high quality and provides significant insight. I recommend the manuscript for publication after the authors have had a chance to address the following comments.

Comments:

In this study it was assumed that the oxidation of SOA by OH occurred only in the

condensed phase. How did the authors rule out the possibility of gas-phase oxidation of semivolatile organics that partition between the condensed-phase and gas-phase after the generation of SOA. This is likely a question that has been addressed in previous studies, but for the uninformed reader, it would be beneficial to discuss in the current manuscript as well.

Page 13, lines 7-9. At this point in the manuscript (without the modelling) the implications to the atmosphere is not completely clear since the reaction time in the laboratory was seconds, while the reaction time in the atmosphere can be days. Because of this difference in reaction times, diffusion limitations can be more important in the laboratory compared to the atmosphere. Hence, at this point in the manuscript, the authors may want to change the wording to "These observations may have implications for the lifetime of SOA with respect to heterogeneous oxidation in the atmosphere".

As pointed out by the authors, mass concentrations of SOA was 1000 microgram/mˆ3, which is higher than ambient concentrations. Can the authors speculate on how their results may change with mass concentration? Two recent studies have explored the effect of mass concentration on the viscosity of SOA. 1,2

References:

(1) Grayson, J. W.; Zhang, Y.; Mutzel, A.; Renbaum-Wolff, L.; Boege, O.; Kamal, S.; Herrmann, H.; Martin, S. T.; Bertram, A. K. Effect of varying experimental conditions on the viscosity of alpha-pinene derived secondary organic material. Atmospheric Chemistry and Physics 2016, 16, 6027-6040. (2) Jain, S.; Fischer, K. B.; Petrucci, G. A. The Influence of Absolute Mass Loading of Secondary Organic Aerosols on Their Phase State. Atmosphere 2018, 9, 14.

---

## Short Comment (SC1) · 3 Aug 2018

We thank the reviewers for the thoughtful comments. We address each comment individually below, with the reviewers' initial comment in **black** and our responses in **blue**.

**Response to Reviewer #1**

The authors explored the heterogeneous oxidation of secondary organic aerosols (SOA) by OH radicals as a function of relative humidity. A difference was observed between heterogeneous oxidation at 25% compared to 80%, and a model was developed to explain the results. This manuscript is easy to follow and clear. The science is also of high quality and provides significant insight. I recommend the manuscript for publication after the authors have had a chance to address the following comments.

Comments:

In this study it was assumed that the oxidation of SOA by OH occurred only in the condensed phase. How did the authors rule out the possibility of gas-phase oxidation of semi-volatile organics that partition between the condensed-phase and gas-phase after the generation of SOA. This is likely a question that has been addressed in previous studies, but for the uninformed reader, it would be beneficial to discuss in the current manuscript as well.

The reviewer raises an important point. After forming the aerosol particles in the first flow tube the air stream is passed through a charcoal denuder to remove the vast majority of gas-phase organic species. It is possible that the SOA evaporates to some extent due to scavenging of the charcoal. However, previous experiments have shown negligible mass loss for SOA simply from scavenging of vapors in denuders on the time scales relevant to our experiment (Cappa and Wilson, 2011).

We already had indicated that a denuder was used, but we have expanded the discussion in Section 2.1 to make clearer the issue raised by the reviewer as:

"Downstream of the flowtube was a Carulite 200 (Carus) denuder followed by a Charcoal denuder to remove residual hydrocarbons and oxidants in the gas phase. *Previous experiments have shown negligible mass loss of SOA particles due to scavenging of vapors in denuders on the time scales relevant to our experiments (Cappa and Wilson, 2011). Therefore, the gas-phase oxidation of semi-volatile organics in the second flowtube is limited.*"

Page 13, lines 7-9. At this point in the manuscript (without the modelling) the implications to the atmosphere is not completely clear since the reaction time in the laboratory was seconds, while the reaction time in the atmosphere can be days. Because of this difference in reaction times, diffusion limitations can be more important in the laboratory compared to the atmosphere. Hence, at this point in the manuscript, the authors may want to change the wording to "These observations may have implications for the lifetime of SOA with respect to heterogeneous oxidation in the atmosphere".

We have made this change.

As pointed out by the authors, mass concentrations of SOA was 1000 microgram/m^3, which is higher than ambient concentrations. Can the authors speculate on how their results may change with mass concentration? Two recent studies have explored the effect of mass concentration on the viscosity of SOA. [1,2]

The reviewer raises an important point. Recent studies explored the effect of mass loadings on the viscosity of SOA under dry conditions (<5%-20% RH) and demonstrated an increase in viscosity with a decrease in mass concentration (Grayson et al., 2016;Jain et al., 2018). While this indicates ambient aerosols are more viscous than lab aerosols generated here under low RH, we focus on the difference of viscosity of SOA between low and high RH. Renbaum-Wolff et al. (2013) have shown that SOA of ambient concentrations (<10 µg m$^{-3}$) are liquid-like ($D_{org}$>10$^{-11}$ cm$^2$ s$^{-1}$) when RH>70%, which is consistent with our model prediction for the $D_{org}$ of SOA of 1000 µg m$^{-3}$. Given that, the difference of viscosity between low and high RH is likely similar or more substantial for ambient SOA than for lab generated SOA. This is because we find that there is a critical point below or above which additional variability in viscosity has only a minor influence on the oxidation behavior. Thus, even if the viscosity of the particles is higher at lower concentrations we would expect to observe similar behavior. Therefore, our results reported here can be applied to smaller mass loadings of SOA and the effect of RH on volume loss and compositional change of aerosols is important to SOA of ambient concentration.

We have added the following discussion in Section 4 to make this point clear.

"Our observations suggest that the lifetime of SOA with respect to heterogeneous OH oxidation is significantly shorter at high RH and that this loss mechanism should be considered in regional and global models. *Although the mass concentrations of SOA we use here are 1 − 3 orders of magnitudes higher than ambient concentrations, the above conclusions are likely to hold true. Recent studies have demonstrated an increase in SOA viscosity with a decrease in mass loadings under low RH conditions (Grayson et al., 2016;Jain et al., 2018). However, under high RH conditions the SOA particles are likely to stay liquid-like independent of mass concentration. Thus, the difference in particle viscosity of several orders of magnitude between low and high RH conditions will almost certainly persist—and perhaps increase—at typical ambient SOA concentrations. Consequently, the influence of RH on SOA aging is likely important even at ambient SOA concentrations.*"

References:

(1) Grayson, J. W.; Zhang, Y.; Mutzel, A.; Renbaum-Wolff, L.; Boege, O.; Kamal, S.; Herrmann, H.; Martin, S. T.; Bertram, A. K. Effect of varying experimental conditions on the viscosity of alpha-pinene derived secondary organic material. Atmospheric Chem- istry and Physics 2016, 16, 6027-6040. (2) Jain, S.; Fischer, K. B.; Petrucci, G. A. The Influence of Absolute Mass Loading of Secondary Organic Aerosols on Their Phase State. Atmosphere 2018, 9, 14.

Cappa, C. D., and Wilson, K. R.: Evolution of organic aerosol mass spectra upon heating: implications for OA phase and partitioning behavior, Atmos Chem Phys, 11, 1895-1911, https://doi.org/10.5194/acp-11-1895-2011, 2011.
Grayson, J. W., Zhang, Y., Mutzel, A., Renbaum-Wolff, L., Boge, O., Kamal, S., Herrmann, H., Martin, S. T., and Bertram, A. K.: Effect of varying experimental conditions on the viscosity of alpha-pinene derived secondary organic material, Atmos Chem Phys, 16, 6027-6040, https://doi.org/10.5194/acp-16-6027-2016, 2016.
Jain, S., Fischer, K. B., and Petrucci, G. A.: The Influence of Absolute Mass Loading of Secondary Organic Aerosols on Their Phase State, Atmosphere-Basel, 9, https://doi.org/10.3390/atmos9040131, 2018.

Renbaum-Wolff, L., Grayson, J. W., Bateman, A. P., Kuwata, M., Sellier, M., Murray, B. J., Shilling, J. E., Martin, S. T., and Bertram, A. K.: Viscosity of alpha-pinene secondary organic material and implications for particle growth and reactivity, P Natl Acad Sci USA, 110, 8014-8019, https://doi.org/10.1073/pnas.1219548110, 2013.

---

## Referee Comment (RC2) · Anonymous Referee #2 · 7 Aug 2018

This study investigated the influence of RH on heterogeneous OH oxidation of a-pinene SOA using laboratory experiments combined with kinetic modeling. They found that SOA evaporated much stronger when oxidized under high RH, which is attributed to RH-dependent differences in the OH uptake coefficient and/or the net probability of fragmentation. The experiments were conducted using an aerosol flow tube and chemical composition was detected using VUV-AMS. I found that experiments were very sophisticated and conducted very well. Application of a kinetic model appears to provide useful insights, but the model does not treat gas diffusion of OH radicals, which appears to be a major limitation of this modeling approach. The manuscript is written overall well and I support publication of this manuscript if the following comments can be addressed and implemented in the revised manuscript.

[Figure]

1. With high uptake coefficients (as OH uptake in this study), it is very important and critical to properly account for gas-phase diffusion for accurate interpretation and analysis of OH uptake (especially if Gamma >0.1). High uptake of OH would lead to depletion of OH in the near-surface gas phase, leading to build up of strong concentration gradients in the gas phase (see textbook such as Seinfeld and Pandis, 2006 etc.). In eq(3), [OH] was used to calculate a collision flux, and I suppose that the author meant average gas-phase OH concentrations with [OH]: this equation would need to be corrected (as OH concentrations close to particles would be depleted) to account for effects of gas-phase diffusion [see, for example, Poschl et al., ACP, 7, 5989, 2010]. Given that the model does not treat gas-phase diffusion, I am not convinced by quantitative conclusions of uptake coefficients (such as 0.2 > Gamma > 0.1, etc.). The lack of gas-phase diffusion treatment appears to be a major weakness of this study (it is not hard to implement this effect, so I encourage authors to implement). If the authors have any evidence that the effects of gas diffusion would be negligible in this study, this should be presented clearly. Without gas diffusion corrections, uptake coefficient you use is not a true uptake coefficient but should be termed as an effective uptake coefficient.

2. Recent studies have shown that ROOH would decompose to form OH and organic radicals under dark [Tong et al., ACP, 16, 1761, 2016] and light conditions [Badali et al., 15, 7831, 2015]. Peroxides contained in a-pinene SOA are shown to be labile [e.g., Epstein et al., EST, 19, 11251, 2014; Krapf et al., Chem, 1, 603, 2016]. In this study, ROOH were treated as stable compounds. I wonder if decomposition of ROOH would have implications on interpretation/analysis of your experiments.

3. Discussion in P23 is very interesting. Slade et al. (GRL, 44, 1583, 2017) observed also very similar that higher generation of heterogeneous oxidation occurs for semisolid particles compared to liquid particles. On a related note, Chim et al. (ACP, 17, 14415, 2017) found that fragmentation and volatilization processes play a larger role than the functionalization process in determining the evolution of aerosol water content and

droplet size at high oxidation stages. These papers may be worth to discuss.

4. How exactly did you calculate model layer thicknesses? By reading L30 in P11, it appears that upon evaporation only the thickness of the sub-surface bulk layers are decreased, but not the thicknesses of the inner bulk, correct?

---

## Author Comment (AC1) · 14 Sep 2018

We thank the reviewers for the thoughtful comments. We address each comment individually below, with the reviewers' initial comment in **black** and our responses in **blue**.

**Response to Reviewer #1**

The authors explored the heterogeneous oxidation of secondary organic aerosols (SOA) by OH radicals as a function of relative humidity. A difference was observed between heterogeneous oxidation at 25% compared to 80%, and a model was developed to explain the results. This manuscript is easy to follow and clear. The science is also of high quality and provides significant insight. I recommend the manuscript for publication after the authors have had a chance to address the following comments.

Comments:

In this study it was assumed that the oxidation of SOA by OH occurred only in the condensed phase. How did the authors rule out the possibility of gas-phase oxidation of semi-volatile organics that partition between the condensed-phase and gas-phase after the generation of SOA. This is likely a question that has been addressed in previous studies, but for the uninformed reader, it would be beneficial to discuss in the current manuscript as well.

The reviewer raises an important point. After forming the aerosol particles in the first flow tube the air stream is passed through a charcoal denuder to remove the vast majority of gas-phase organic species. It is possible that the SOA evaporates to some extent due to scavenging of the charcoal. However, previous experiments have shown negligible mass loss for SOA simply from scavenging of vapors in denuders on the time scales relevant to our experiment (Cappa and Wilson, 2011).

We already had indicated that a denuder was used, but we have expanded the discussion in Section 2.1 to make clearer the issue raised by the reviewer as:

"Downstream of the flowtube was a Carulite 200 (Carus) denuder followed by a Charcoal denuder to remove residual hydrocarbons and oxidants in the gas phase. *Previous experiments have shown negligible mass loss of SOA particles due to scavenging of vapors in denuders on the time scales relevant to our experiments (Cappa and Wilson, 2011). Therefore, the gas-phase oxidation of evaporated semi-volatile organics in the second flowtube is likely limited.*"

Page 13, lines 7-9. At this point in the manuscript (without the modelling) the implications to the atmosphere is not completely clear since the reaction time in the laboratory was seconds, while the reaction time in the atmosphere can be days. Because of this difference in reaction times, diffusion limitations can be more important in the laboratory compared to the atmosphere. Hence, at this point in the manuscript, the authors may want to change the wording to "These observations may have implications for the lifetime of SOA with respect to heterogeneous oxidation in the atmosphere".

We have made this change.

As pointed out by the authors, mass concentrations of SOA was 1000 microgram/mˆ3, which is higher than ambient concentrations. Can the authors speculate on how their results may change with mass concentration? Two recent studies have explored the effect of mass concentration on the viscosity of SOA.

The reviewer raises an important point. Recent studies explored the effect of mass loadings on the viscosity of SOA under dry conditions (<5%-20% RH) and demonstrated an increase in viscosity with a decrease in mass concentration (Grayson et al., 2016;Jain et al., 2018). While this indicates ambient aerosols are more viscous than lab aerosols generated here under low RH, we focus on the difference of viscosity of SOA between low and high RH. Renbaum-Wolff et al. (2013) have shown that SOA of ambient concentrations (<10 µg m$^{-3}$) are liquid-like ($D_{org}$>10$^{-11}$ cm$^2$ s$^{-1}$) when RH>70%, which is consistent with our model prediction for the $D_{org}$ of SOA of 1000 µg m$^{-3}$. Given that, the difference of viscosity between low and high RH is likely similar or more substantial for ambient SOA than for lab generated SOA. This is because we find that there is a critical point below or above which additional variability in viscosity has only a minor influence on the oxidation behavior. Thus, even if the viscosity of the particles is higher at lower concentrations we would expect to observe similar behavior. Therefore, our results reported here can be applied to smaller mass loadings of SOA and the effect of RH on volume loss and compositional change of aerosols is important to SOA of ambient concentration.

We have added the following discussion in Section 4 to make this point clear.

"Our observations suggest that the lifetime of SOA with respect to heterogeneous OH oxidation is significantly shorter at high RH and that this loss mechanism should be considered in regional and global models. *Although the mass concentrations of SOA we use here are 1 – 3 orders of magnitudes higher than ambient concentrations, the above conclusions are likely to hold true. Recent studies have demonstrated an increase in SOA viscosity with a decrease in mass loadings under low RH conditions (Grayson et al., 2016;Jain et al., 2018). However, under high RH conditions the SOA particles are likely to stay liquid-like independent of mass concentration. Thus, the difference in particle viscosity of several orders of magnitude between low and high RH conditions will almost certainly persist—and perhaps increase—at typical ambient SOA concentrations. Consequently, the influence of RH on SOA aging is likely important even at ambient SOA concentrations.*"

**References:**

Cappa, C. D., and Wilson, K. R.: Evolution of organic aerosol mass spectra upon heating: implications for OA phase and partitioning behavior, Atmos Chem Phys, 11, 1895-1911, https://doi.org/10.5194/acp-11-1895-2011, 2011.

Grayson, J. W., Zhang, Y., Mutzel, A., Renbaum-Wolff, L., Boge, O., Kamal, S., Herrmann, H., Martin, S. T., and Bertram, A. K.: Effect of varying experimental conditions on the viscosity of alpha-pinene derived secondary organic material, Atmos Chem Phys, 16, 6027-6040, https://doi.org/10.5194/acp-16-6027-2016, 2016.

Jain, S., Fischer, K. B., and Petrucci, G. A.: The Influence of Absolute Mass Loading of Secondary Organic Aerosols on Their Phase State, Atmosphere-Basel, 9, https://doi.org/10.3390/atmos9040131, 2018.

Renbaum-Wolff, L., Grayson, J. W., Bateman, A. P., Kuwata, M., Sellier, M., Murray, B. J., Shilling, J. E., Martin, S. T., and Bertram, A. K.: Viscosity of alpha-pinene secondary organic material and implications for particle growth and reactivity, P Natl Acad Sci USA, 110, 8014-8019, https://doi.org/10.1073/pnas.1219548110, 2013.

**Response to Reviewer #2**

This study investigated the influence of RH on heterogeneous OH oxidation of a-pinene SOA using laboratory experiments combined with kinetic modeling. They found that SOA evaporated much stronger when oxidized under high RH, which is attributed to RH-dependent differences in the OH uptake coefficient and/or the net probability of fragmentation. The experiments were conducted using an aerosol flow tube and chemical composition was detected using VUV-AMS. I found that experiments were very sophisticated and conducted very well. Application of a kinetic model appears to provide useful insights, but the model does not treat gas diffusion of OH radicals, which appears to be a major limitation of this modeling approach. The manuscript is written overall well and I support publication of this manuscript if the following comments can be addressed and implemented in the revised manuscript.

Comments:

1. With high uptake coefficients (as OH uptake in this study), it is very important and critical to properly account for gas-phase diffusion for accurate interpretation and analysis of OH uptake (especially if Gamma >0.1). High uptake of OH would lead to depletion of OH in the near-surface gas phase, leading to build up of strong concentration gradients in the gas phase (see textbook such as Seinfeld and Pandis, 2006 etc.). In eq(3), [OH] was used to calculate a collision flux, and I suppose that the author meant average gas-phase OH concentrations with [OH]: this equation would need to be corrected (as OH concentrations close to particles would be depleted) to account for effects of gas-phase diffusion [see, for example, Poschl et al., ACP, 7, 5989, 2010]. Given that the model does not treat gas-phase diffusion, I am not convinced by quantitative conclusions of uptake coefficients (such as 0.2 > Gamma > 0.1, etc.). The lack of gas-phase diffusion treatment appears to be a major weakness of this study (it is not hard to implement this effect, so I encourage authors to implement). If the authors have any evidence that the effects of gas diffusion would be negligible in this study, this should be presented clearly. Without gas diffusion corrections, uptake coefficient you use is not a true uptake coefficient but should be termed as an effective uptake coefficient.

The reviewer raises an important point. The effect of gas diffusion and OH depletion near surface due to high OH uptake is indeed an important aspect. This effect was actually considered in the model according to Equation (9) and (19) in Pöschl et al. (2007) but we failed to explicitly state that this was done in the manuscript. Therefore, the following description of gas diffusion correction and equations have been added to Section 2.5:

"$J_{coll}$ is the OH radicals flux at the particle surface, calculated as:

$$J_{coll} = \bar{c} \cdot [OH]_{gs}/4 \qquad\qquad (3)$$

where $\bar{c}$ is the mean speed of gas-phase OH *and $[OH]_{gs}$ is the near-surface concentration of gas-phase OH. The relationship between $[OH]_{gs}$ and $[OH]_g$, the concentration far from the particle surface, can be described as:*

$$[OH]_{gs} = C_{g,OH}[OH]_g \tag{4}$$

*where $C_{g,OH}$ is a gas-phase diffusion correction factor suggested by Pöschl et al. (2007):*

$$C_{g,OH} = \frac{1}{1+\gamma_{OH}^Y \frac{0.75+0.28\,Kn_{OH}}{Kn_{OH}(1+Kn_{OH})}} \tag{5}$$

*$Kn_{OH}$ is the Knudsen number, approximated as:*

$$Kn_{OH} = \frac{6\,D_{g,OH}}{\bar{c}\,D_{p,S}} \tag{6}$$

*where $D_{g,OH}$ is the OH gas phase diffusion coefficient and $D_{p,S}$ is the surface-weighted particle diameter. This correction is needed to account for gas-phase diffusion especially when high uptake of gas-phase OH leads to strong local depletion of OH near particle surface."*

2. Recent studies have shown that ROOH would decompose to form OH and organic radicals under dark [Tong et al., ACP, 16, 1761, 2016] and light conditions [Badali et al., 15, 7831, 2015]. Peroxides contained in a-pinene SOA are shown to be labile [e.g., Epstein et al., EST, 19, 11251, 2014; Krapf et al., Chem, 1, 603, 2016]. In this study, ROOH were treated as stable compounds. I wonder if decomposition of ROOH would have implications on interpretation/analysis of your experiments.

The reviewer raises an interesting point. Organic peroxides have been observed to contribute significantly to $\alpha$-pinene+$O_3$ SOA (as much as 50%) and are thermally unstable, with half-lives estimated at less than an hour at room temperature for some organic peroxides. For the timescale relevant to our experiment, 38 s, decomposition of peroxides is unlikely to drive difference between the mass loss of solid and liquid-like aerosol particles. However, under ambient conditions and longer timescales, decomposition of peroxides might make an additional contribution to aerosol mass loss through the formation of condensed-phase OH radicals and RO radicals that lead to fragmentation, especially under high RH conditions with the presence of liquid water in the particles. We now address this point in Section 3.4.2:

*"OH radicals can also be produced in the condensed phase through decomposition of organic peroxides (ROOH). ROOH molecules can contribute as much as 50% to the total mass of $\alpha$-pinene+$O_3$ SOA (Docherty et al., 2005) but are unstable and with some observed to have half-lives with respect to decomposition of*

*less than an hour at room temperature (Krapf et al., 2016). The decomposition products of ROOH—RO and OH radicals—both facilitate the formation of small molecular weight products that lead to volatilization. Although this chemical pathway is unlikely to contribute substantially in our experiments given the short experiment timescale (38 s), it may influence the rate of aerosol mass loss under ambient conditions and longer timescales, especially under high RH conditions where appreciable amount of liquid water in the particles accelerates the decomposition of ROOH (Tong et al., 2016). "*

3. Discussion in P23 is very interesting. Slade et al. (GRL, 44, 1583, 2017) observed also very similar that higher generation of heterogeneous oxidation occurs for semisolid particles compared to liquid particles. On a related note, Chim et al. (ACP, 17, 14415, 2017) found that fragmentation and volatilization processes play a larger role than the functionalization process in determining the evolution of aerosol water content and droplet size at high oxidation stages. These papers may be worth to discuss.

We thank the reviewer for providing more insights into the discussion. We have expanded the discussion in two ways. First, in Section 3.3.1 we have added discussion in relation to Slade et al. (2017). Here, we focused on what they concluded regarding how the surface vs. the bulk was oxidized. We did not, at this point, comment specifically on the implications regarding particle hygroscopicity.

Second, we added very brief discussion in the Conclusions related to the potential importance of this work for particle hygroscopicity, using the Slade et al. (2017) and Chim et al. (2017) results as motivation.

*"Our observations and modeling indicate that the phase state affects the distribution and oxidation level of the organic species in the particles upon photochemical aging. Slade et al. (2017) oxidized Suwannee River fulvic acid (SRFA) particles above and below the glass transition temperature ($T_g$) up to OH exposures of 7 x 10$^{11}$ molecule cm$^{-3}$ s$^{-1}$, an order or magnitude lower than our highest value. Above $T_g$ the $D_{org}$ is much larger than below the $T_g$, with an approximate transition from highly viscous to liquid like around 295-300 K. As such, these T-dependent experiments are complementary to the RH-dependent experiments here. While they did not measure the particle composition, using the KM-GAP model (Shiraiwa et al., 2010;Shiraiwa et al., 2012) they concluded that oxidation at low temperatures (<295 K; semi-solid) engendered compositional changes primarily at the particle surface, while at high temperatures (>300 K; liquid like) changes occurred throughout the particle, consistent with our results. Consequently, for the low-T experiments they predicted that the extent of oxidation of molecules in the near-surface layers, specifically, was much greater than occurred throughout the entire particle in the high-T experiments, also consistent with our conclusions. Associated, we find that the concentration of highly oxidized (3$^{rd}$ or higher generation) species in the surface layer rapidly increases in the low-RH (low $D_{org}$) simulations (**Figure S3**). In contrast, for the high RH simulations the concentration of highly oxidized molecules increases only slowly, but continuously, and ultimately reaches a higher concentration than for the low RH simulations (**Figure S3**). However, we note that these highly oxidized molecules in our low-RH experiments make up only a small fraction of the total molecules comprising the particle (**Figure 10** and **Figure S3**), which is somewhat inconsistent with the conclusion of Slade et al. (2017) that large changes to the average molecular weight of the organic species comprising the particle occur."*

*"Particle phase-dependent differences in oxidation—including the importance of functionalization of fragmentation at the surface or throughout the bulk of a particle—can impact how the hygroscopicity of organic particles evolves (Chim et al., 2017;Slade et al., 2017)."*

[Figure]

*Figure S3. Variation in the concentration of the parent species (filled symbols) and 3rd and higher generation species (open symbols) in the surface layer for low-RH/small-$D_{org}$ simulations (red lines/points) and for the high-RH/large-$D_{org}$ simulations (blue lines/points). These are example results for simulations assuming $\gamma = 0.5$, $p_{frag} = 0.31$, GF = 1.0 and $k_{RO2+RO2} = 1 \times 10^{-15}$ cm$^3$ molecule$^{-1}$ s$^{-1}$ (same conditions as Figure 10).*

4. How exactly did you calculate model layer thicknesses? By reading L30 in P11, it appears that upon evaporation only the thickness of the sub-surface bulk layers are decreased, but not the thicknesses of the inner bulk, correct?

The thickness (radius) of the inner bulk is set equal to the thickness of sub-surface bulk layers and is treated as a "sub-surface layer". For example, if the particle has a radius of 60.00 nm, the depth of surface layer is set to 0.76 nm constant, and total layer number is 50, then the thickness of the sub-surface layers and the radius of the inner core are both (60.00 - 0.76)/50 = 1.18 nm. We have revised the statement as following:

"As particles shrink, the thickness of the sub-surface bulk layers *(including the radius of the inner core)* decreases based on the new particle size and constant total layer number, …."

**References**

Chim, M. M., Cheng, C. T., Davies, J. F., Berkemeier, T., Shiraiwa, M., Zuend, A., and Chan, M. N.: Compositional evolution of particle-phase reaction products and water in the heterogeneous OH oxidation of model aqueous organic aerosols, Atmos Chem Phys, 17, 14415-14431, 10.5194/acp-17-14415-2017, 2017.

Docherty, K. S., Wu, W., Lim, Y. B., and Ziemann, P. J.: Contributions of organic peroxides to secondary aerosol formed from reactions of monoterpenes with O-3, Environmental Science & Technology, 39, 4049-4059, 2005.

Krapf, M., El Haddad, I., Bruns, E. A., Molteni, U., Daellenbach, K. R., Prevot, A. S. H., Baltensperger, U., and Dommen, J.: Labile Peroxides in Secondary Organic Aerosol, Chem-Us, 1, 603-616, https://doi.org/10.1016/j.chempr.2016.09.007, 2016.

Pöschl, U., Rudich, Y., and Ammann, M.: Kinetic model framework for aerosol and cloud surface chemistry and gas-particle interactions - Part 1: General equations, parameters, and terminology, Atmos Chem Phys, 7, 5989-6023, 10.5194/acp-7-5989-2007, 2007.

Shiraiwa, M., Pfrang, C., and Poschl, U.: Kinetic multi-layer model of aerosol surface and bulk chemistry (KM-SUB): the influence of interfacial transport and bulk diffusion on the oxidation of oleic acid by ozone, Atmos Chem Phys, 10, 3673-3691, https://doi.org/10.5194/acp-10-3673-2010, 2010.

Shiraiwa, M., Pfrang, C., Koop, T., and Poschl, U.: Kinetic multi-layer model of gas-particle interactions in aerosols and clouds (KM-GAP): linking condensation, evaporation and chemical reactions of organics, oxidants and water, Atmos Chem Phys, 12, 2777-2794, https://doi.org/10.5194/acp-12-2777-2012, 2012.

Slade, J. H., Shiraiwa, M., Arangio, A., Su, H., Poschl, U., Wang, J., and Knopf, D. A.: Cloud droplet activation through oxidation of organic aerosol influenced by temperature and particle phase state, Geophysical Research Letters, 44, 1583-1591, https://doi.org/10.1002/2016GL072424, 2017.

Tong, H. J., Arangio, A. M., Lakey, P. S. J., Berkemeier, T., Liu, F. B., Kampf, C. J., Brune, W. H., Poschl, U., and Shiraiwa, M.: Hydroxyl radicals from secondary organic aerosol decomposition in water, Atmos Chem Phys, 16, 1761-1771, https://doi.org/10.5194/acp-16-1761-2016, 2016.